# splitSMLM, a spectral demixing method for high-precision multi-color localization microscopy applied to nuclear pore complexes

Leonid Andronov [1,2,3,4,5], Rachel Genthial[1,2,3,4,5], Didier Hentsch[1,2,3,4,5] & Bruno P. Klaholz [1,2,3,4,5✉]

Single molecule localization microscopy (SMLM) with a dichroic image splitter can provide invaluable multi-color information regarding colocalization of individual molecules, but it often suffers from technical limitations. Classical demixing algorithms tend to give suboptimal results in terms of localization precision and correction of chromatic errors. Here we present an image splitter based multi-color SMLM method (splitSMLM) that offers much improved localization precision and drift correction, compensation of chromatic distortions, and optimized performance of fluorophores in a specific buffer to equalize their reactivation rates for simultaneous imaging. A novel spectral demixing algorithm, SplitViSu, fully preserves localization precision with essentially no data loss and corrects chromatic errors at the nanometer scale. Multi-color performance is further improved by using optimized fluorophore and filter combinations. Applied to three-color imaging of the nuclear pore complex (NPC), this method provides a refined positioning of the individual NPC proteins and reveals that Pom121 clusters act as NPC deposition loci, hence illustrating strength and general applicability of the method.

[1] Centre for Integrative Biology (CBI), Department of Integrated Structural Biology, IGBMC, CNRS, Inserm, Université de Strasbourg, 1 rue Laurent Fries, 67404 Illkirch, France. [2] Institute of Genetics and of Molecular and Cellular Biology (IGBMC), 1 rue Laurent Fries, Illkirch, France. [3] Centre National de la Recherche Scientifique (CNRS), UMR 7104 Illkirch, France. [4] Institut National de la Santé et de la Recherche Médicale (Inserm), U964 Illkirch, France. [5] Université de Strasbourg, Illkirch, France. ✉email: klaholz@igbmc.fr

uper-resolution microscopy breaks the diffraction limit of resolution in fluorescence microscopy and hence allows bioimaging with unprecedented details. Among the numerous super-resolution imaging techniques, stimulated emission depletion (STED) and structured illumination microscopy (SIM) rely on special illumination schemes[1,2]. Single-molecule localization microscopy (SMLM) techniques include stochastic optical reconstruction microscopy (STORM), photo-activated localization microscopy (PALM) and point accumulation for imaging in nanoscale topography (PAINT). They use rather conventional epifluorescence schemes and rely on specific photophysical and photochemical properties of dyes that are repeatedly switched on and off, imaging only a sparse subset of molecules at a time[3–5]. MINFLUX and similar techniques combine both approaches and localize individually switched-on fluorophores with a patterned illumination[6–9]. SMLM is one of the most heavily used bioimaging super-resolution techniques because it employs conventional optics and simple labeling[10] and yet provides excellent resolution in the 10–20 nm range, usually higher than what is achievable with STED and much higher than with SIM, hence allowing imaging at the molecular level.

An important advantage of fluorescence imaging is its multi-color capability. This allows for colocalization of objects through simultaneous imaging of different targets in a given sample. While classical colocalization in confocal microscopy is limited in resolution, super-resolution microscopy combined with multi-color imaging can help deciphering interactions between proteins at the single-molecule level[11–13]. Multi-color imaging is possible in SIM and STED using fluorophores with substantially different spectral properties[14] or lifetimes[15]. SMLM, however, opens additional possibilities for multi-color imaging, thanks to the access to the fluorescence properties of individual molecules, enabling separation of fluorophores with very close spectra using a dichroic image splitter and ratiometry[16] or single-molecule spectrometry[17]. Potential advantages of simultaneous multi-color imaging with spectrally close fluorophores include: (1) several species of labels are acquired simultaneously, improving the imaging speed; (2) thanks to simultaneous imaging, drift is the same in all channels and can be corrected more reliably; (3) spectrally close fluorophores often have similar photophysical properties and can provide equivalent data quality for different color channels; (4) non-reliable localizations originating from autofluorescence or noise can be filtered out[18,19]; (5) chromatic errors can be avoided; (6) the excitation path of the microscope is simplified because only one excitation laser is necessary. However, while the potential of multi-color SMLM is strong, only few implementations actually allow to do this conveniently and precisely.

The currently available software for demixing[18,20] implement a rather simple algorithm to identify localization pairs from two spectral channels, followed by fluorophore assignment based on their intensities within the channels, and listing of demixed coordinates using the coordinates of one of the input channels. Previous implementations of multi-color SMLM with an image splitter are limited by photon budget usage that decreases localization precision[18]. Setups that can detect "salvaged fluorescence" (i.e., within a narrow spectral region close to the excitation laser wavelength) are very complex and require two objectives and rather inconvenient sample mounting between two coverslips[21]. Additionally, the performance of simultaneously imaged fluorophores is not equivalent, which limits the range of attainable imaging parameters, e.g. the excitation intensity cannot be lowered in order to improve resolution[22]. To help overcoming some of these limitations we designed a newly optimized approach.

In this work, we describe a simple and robust implementation of multi-color SMLM with a spectral image splitter (splitSMLM)

resulting in much improved localization precision, drift correction, compensation of chromatic errors and enhanced performance of fluorophores in optimized buffers. Our demixing algorithm uses all detected photons for both spectral separation and localization, which improves image resolution. Our finely selected filters allow detection of salvaged fluorescence on a conventional single-objective setup while our optimized imaging buffer improves the photon budget of fluorophores and equalizes their response on reactivating light.

To implement the spectral demixing data processing for multi-color SMLM, we developed a software, SplitViSu, which is freely available and is integrated into a workflow of image analysis, reconstruction and segmentation of SMLM data within the SharpViSu platform[23]. We demonstrate the performance of our methods by imaging the nuclear pore complex (NPC), a reference standard for super-resolution microscopy[24], revealing simultaneous localization of individual Nup62, Nup96, Pom121 and lamin B1 proteins within the nuclear envelope and providing new findings on their colocalization.

## Results

**Design**. Separating fluorophores with similar spectra can be achieved with a dichroic mirror that divides the imaging path of the microscope into two channels, the short wavelength ($\lambda_S$) channel and the long wavelength ($\lambda_L$) channel, followed by simultaneous recording of the two channels on the camera[16,25–27] (Fig. 1, Supplementary Fig. 1). We implemented this with minor modifications of the microscope, using a commercial image splitter Optosplit II (Cairn Research) (alternatively, a W-View Gemini, Hamamatsu Photonics, could be used), where the two channels are imaged side-by-side on the same camera chip (Fig. 1a, c). The coordinates ($X_S$, $X_L$) and the brightness of the fluorophores ($I_S$, $I_L$) within the split camera image can be determined using a conventional localization algorithm[28], while additional processing is needed to assign the localizations to the fluorophores.

For each fluorophore, the ratio of its photon counts in the two channels is

$$r = I_L/I_S \qquad (1)$$

This ratio will depend on the emission spectra of the fluorophore and on the transmission (reflection) spectrum of the dichroic mirror, providing different ratios for fluorophores even with only slightly different emission spectra (Fig. 1d, e). For spectral demixing, we use a bivariate histogram of intensities in the $\lambda_L$ and $\lambda_S$ channels (Fig. 1d, Supplementary Fig. 2a–c). On such a histogram, the localizations originating from a given fluorophore are distributed within a sector oriented along a line with slope $r$ and zero y-intercept:

$$I_L = r \cdot I_S \qquad (2)$$

or if represented on a log-log graph (Supplementary Fig. 2c), along a line with slope 1 and y-intercept $log(r)$:

$$log(I_L) = log(r) + log(I_S) \qquad (3)$$

The non-selected points outside of the sector are rejected and are not used for image reconstruction. The bivariate histogram has an advantage over a univariate histogram of ratios $r$ (Fig. 1e, Supplementary Fig. 2d) because it allows demixing or filtration based not only on ratios but also on absolute values of intensities. As spectral demixing becomes less reliable at low intensities, a bivariate histogram allows for removal of low-intensity localizations that cannot be reliably assigned to a fluorophore.

Among the best fluorophores for SMLM so far are the molecules based on the cyanine scaffold with emission in the far-red and near-infrared spectral ranges of 620–750 nm[29]. These include

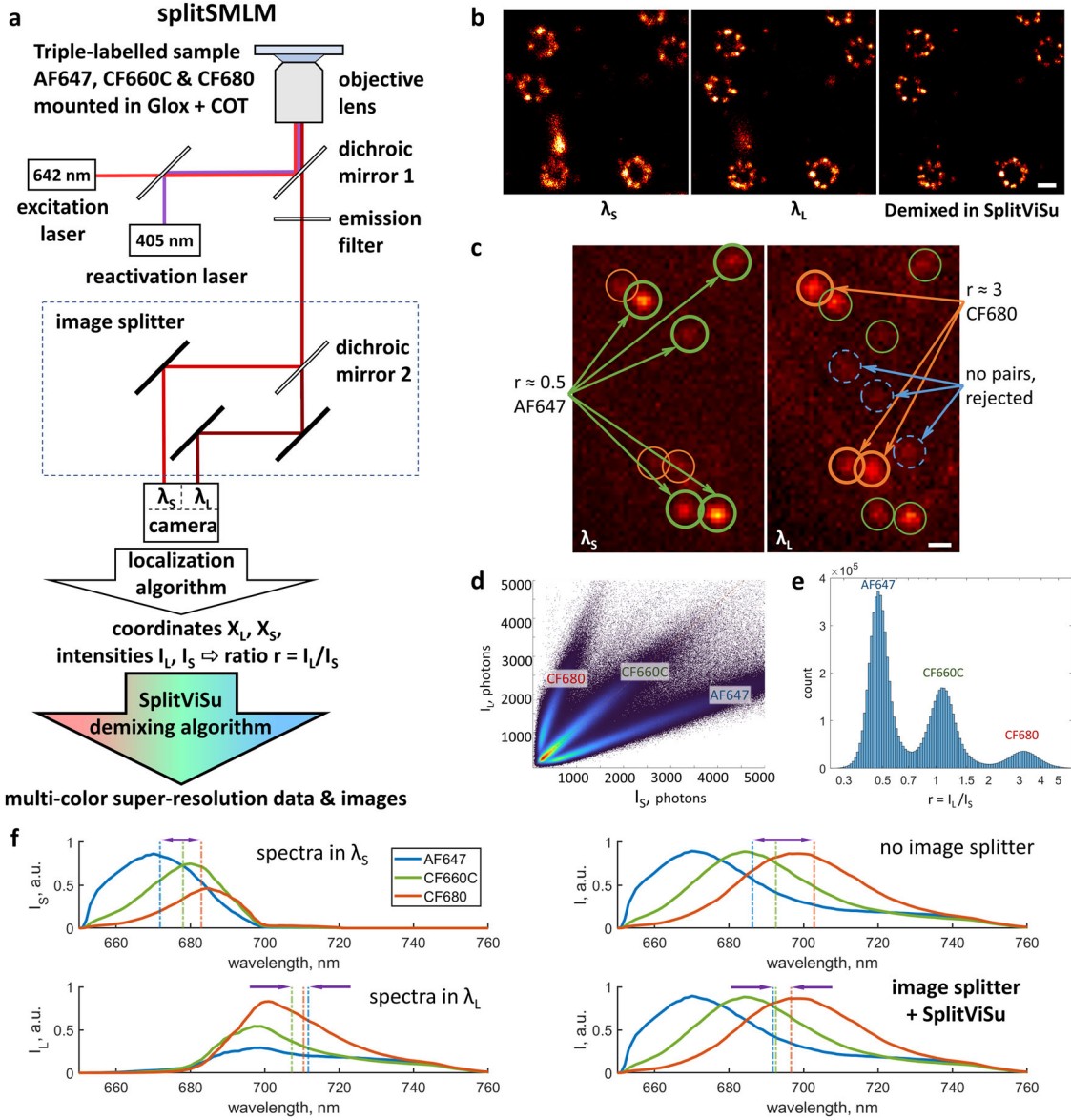

**Fig. 1 Principle of splitSMLM. a** Scheme of the splitSMLM imaging system and the overall workflow. **b** Super-resolution images of a single-labeled sample, reconstructed (from left to right) from localizations within the $\lambda_S$ channel, from localizations within the $\lambda_L$ channel or after demixing in SplitViSu, demonstrating removal of spurious localizations and resolution improvement. **c** Image of the two spectrally different channels, $\lambda_S$ and $\lambda_L$ (left and right, respectively), with localizations originating either from AF647 (green), CF680 (orange) or from background noise (blue). **d** Bivariate histogram of photon counts $I_S$ and $I_L$ originating from the $\lambda_S$ and $\lambda_L$ channels with three sectors corresponding to three fluorophore species. **e** Univariate histogram of ratios $r$ on a semi-log plot with three peaks corresponding to three fluorophore species. **f** Fluorescence spectra of AF647, CF660C and CF680 calculated within the $\lambda_S$ and $\lambda_L$ channels (left), without image splitter (top right), or equivalent spectra after demixing in SplitViSu (bottom right). Vertical lines represent average wavelengths for the corresponding fluorophores in the corresponding channels. Average wavelengths after demixing in SplitViSu are calculated as the mean values of the average wavelengths of the fluorophores in the $\lambda_S$ and $\lambda_L$ channels. Arrows indicate maximum spectral difference within the triplet of fluorophores and reflect the amount of chromatic aberrations in the corresponding setting. Scale bars, 100 nm (**b**) and 500 nm (**c**).

Alexa Fluor 647 (AF647), CF680, CF660C, DY643, DY654, DyLight 680[21,22,29]. Their close excitation and emission spectra do not allow conventional multi-color imaging where each dye is excited with a separate laser and is detected through a separate emission filter. However, they are perfectly suited for excitation with a single laser and for simultaneous imaging when their emission is split into two channels with a dichroic splitter (Fig. 1f, Supplementary Fig. 1b). Among those fluorophores, we chose a triplet of AF647, CF660C and CF680[30], because these three dyes have regularly spaced emission spectra and are widely available as antibody conjugates. Additionally, they have already been characterized and perform well for multi-color SMLM in the

classical Glucose Oxidase—Catalase buffer with addition of β-mercaptoethylamine (MEA) or β-mercaptoethanol[22].

**Maximizing the localization precision and minimizing chromatic errors in multi-color imaging.** The precision of single-molecule localization increases with the square root of the number of detected photons[31]. In order to achieve the highest resolution, it is therefore important to detect as many photons as possible. At the same time, the dichroic mirror in the image splitter strongly increases the spectral shift between fluorophores detected in different channels (Fig. 1f), which might increase lateral chromatic aberrations. For this reason, using an image

**Fig. 2 Localization precision and chromatic error correction in splitSMLM using SplitViSu. a** Simulated results of demixing in SplitViSu using various methods for calculation of the output coordinates: "brightest"—the output coordinates equal the input coordinates from the brightest channel for the given fluorophore; "$\lambda_S$", "$\lambda_L$"—the output coordinates equal the input coordinates from the $\lambda_S$ or $\lambda_L$ channel, respectively; "simple mean"—the output is calculated as a simple mean of the input coordinates in the $\lambda_S$ and $\lambda_L$ channels; "weighted mean"—the output is calculated as a photon count-weighted mean of the input coordinates in the $\lambda_S$ and $\lambda_L$ channels; "wmean-chroma"—the output is calculated as the "weighted mean" with subsequent subtraction of a chromatic error. The simulated data contains two species of fluorophores with different ratios: $r_{red} = 0.6$, $r_{blue} = 3.5$. The true position is the same for both fluorophore species and is shown as a black dot. The total photon count detected from every localization is the same for all fluorophores and equals 2000 photons. The full width at half maximum (FWHM) of the microscope's PSF is 300 nm. The chromatic error between two fluorophores within the opposite channels of the image splitter is 24 nm (x-direction) and 8 nm (y-direction). Each species was detected 50 times within each channel. The circles represent the FWHM of the demixed distributions using the corresponding method for calculation of the output coordinates. **b** Scheme of the NPCs at the NE, used as a test object in this study. ONM, outer nuclear membrane; INM, inner nuclear membrane. **c–e** "Top views" of the NPCs in a U2OS cell with immunofluorescently labeled Pom121 (blue), Nup62 (red) and Nup96 (green). Rectangle in **c** represents the region zoomed in **d**; square in **d** represents the region zoomed in **e**. Circles in **c** and **e** represent Pom121 clusters with few localizations of Nup96 and Nup62, which might correspond to deposition sites of new NPCs. **e** Different methods for calculation of demixed coordinates tested on a single NPC. Numbers in the bottom represent the resolution of the images, estimated according to the FRC$_{1/7}$ criterion[33], calculated from the images of a whole cell. The images in **e** are reconstructed as 2D histograms of localization coordinates with a pixel size of 5 × 5 nm. **f** Sum of aligned images of individual NPCs with a radial distribution of localizations of each protein shown in the graph. For this graph, a sum of 160 NPC particles was used. Scale bars, 1 μm (**c**), 200 nm (**d**), 50 nm (**e**, **f**).

splitter, the coordinates of demixed localizations are commonly calculated from only one spectral channel[21,32]. However, this approach leaves the photons in the second channel unused and therefore reduces the localization precision; the photons in the second channel are utilized only to calculate the ratio of photons between two channels. Because the signal of the same channel of the image splitter is used for image reconstruction in two or three different colors, this approach reduces chromatic errors (Fig. 1f, Supplementary Fig. 3).

To improve the localization precision, it is possible to widen the spectral width of the channel used for localization (generally $\lambda_L$) and to narrow that of the other channel used only for demixing (generally $\lambda_S$)[9,21]. This tends to decrease the signal within $\lambda_S$ (that hence becomes the narrow-band channel), leading to an under-detection of fluorophores in the $\lambda_S$ channel, especially the species with longer-wavelength emission spectra. For fluorophores with emission in the shorter wavelength range, a substantial number of photons will be detected in the $\lambda_S$ channel but will not be utilized for localization. In splitSMLM, we use $\lambda_S$ and $\lambda_L$ channels of similar spectral width that provides similar intensities of fluorescence and background in both $\lambda_S$ and $\lambda_L$, which allows robust localization with any fitting algorithm using the same parameters for both channels.

In our setup, we aimed at improving the localization precision by including the signals from both channels into the computation of the coordinates of the demixed fluorophores. We calculate the output demixed coordinates as a weighted mean of the input coordinates in the two spectral channels insuring best localization precision. Then we remove any residual chromatic distortion using the simple mean of the input coordinates as an aberration-free reference (see Methods, Supplementary Fig. 4). The resulting localizations have the best possible resolution, as determined by Fourier ring correlation (FRC)[33] (Fig. 2a, Supplementary Fig. 5). Positive control for chromatic error correction using different labels on the same antibodies reveals colocalization, the extent of which may be limited by site accessibility by related antibodies (Supplementary Fig. 5).

**Cross-talk between demixed channels**. Due to various imperfections, such as errors in estimation of photon counts, noise, emission of several molecules within a single diffraction-limited region, variations in the emission spectra of individual molecules[34], some localizations of one fluorophore species can adopt $r$-values that approach that of another species (Supplementary Fig. 6c–e). This leads to a cross-talk between the channels, i.e., to appearance of localizations of a different fluorophore

within a demixed channel. It is therefore important to carefully choose the molecules within a sector on the histogram of photon counts in order to minimize both rejection of fluorophores and cross-talk using single-labeled samples (Supplementary Fig. 6, 7b). The intersection between the experimental spread and the applied sector region of the other color describes the level of cross-talk.

Typically, the cross-talk is estimated by comparing the number of demixed localizations between different output channels using single-labeled samples[16]. In our setup, the cross-talk determined by this method is <2% for any pair of fluorophores within broad regions of bivariate histogram that insure low (about 10%) rejection of localizations (Supplementary Fig. 6, 7). Also, some dim localizations can be detected even within a non-labeled sample (Supplementary Fig. 6f), which can lead to overestimation of cross-talk (hence cross-talk is even lower, see Supplementary Figs. 6, 7b). Importantly, no cross-talk can be visually observed between demixed channels in our reconstructed images (Figs. 2–4).

**Refinement of multiple localizations for SMLM image reconstruction**. For SMLM image reconstruction, we developed a method for processing of re-localizations of molecules on consecutive frames. Usually, consecutive localizations are merged into a single value that is located at the mean position of the initial localizations. This improves their precision but reduces the density of localizations that might not necessarily result in resolution improvement (Supplementary Figs. 8, 9). In our method, instead of being reduced to only one localization, all initial localizations are kept, but their coordinates are refined. The new coordinates are selected randomly from a normal distribution with a standard deviation

$$\sigma = \sigma_{psf} / \sqrt{N_{ph}} \qquad (4)$$

where $\sigma_{psf}$ is the standard deviation of the point spread function of the microscope ($\sigma_{psf} = 140$ nm in our setup) and $N_{ph}$ is the sum of the photon counts of the series of consecutive localizations. This is done to reflect new (i.e., improved) localization precision and to avoid appearance of very bright pixel artefacts in the reconstructed images, which would happen if all localizations of the group were set to have exactly the same coordinates[28]. The center of the refined distribution lies at the weighted mean position of the initial localizations, the weights of the initial localizations being proportional to their photon counts. This method improves the resolution of the reconstructed images, both visually and according to the FRC criterion[33,35] (Fig. 2e,

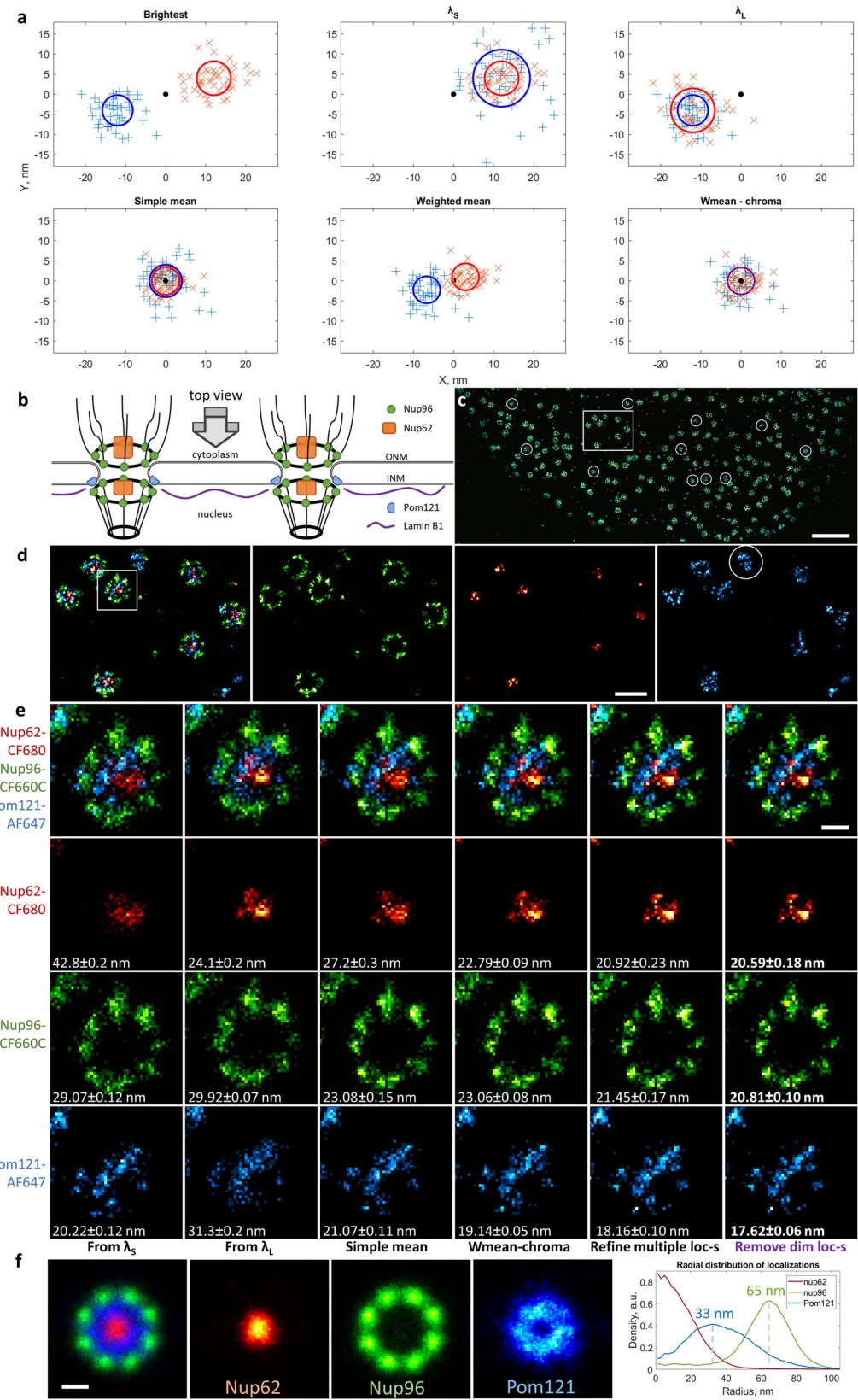

Supplementary Figs. 8, 9). After the refinement, the localizations with low photon numbers can be additionally removed, which further improves resolution (Fig. 2e). This method of processing of consecutive localizations is now implemented in the software suite SharpViSu[23].

**SplitViSu, a software for demixing of spectrally close fluorophores**. To integrate different aspects of fluorophore colocalization analysis into a convenient tool, we developed SplitViSu, a software for demixing SMLM data acquired with an image splitter (Supplementary Fig. 10). The software has an interactive

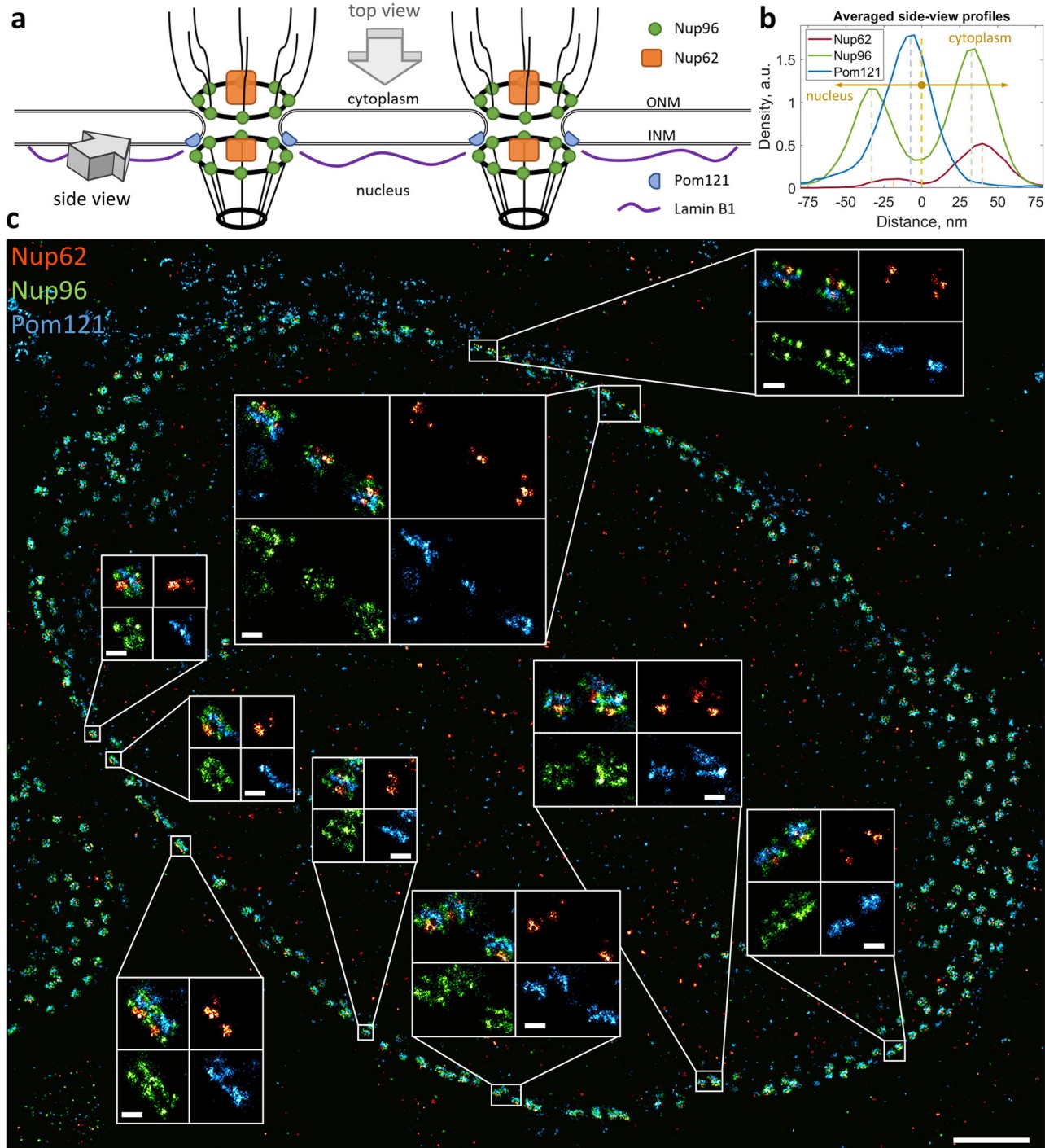

**Fig. 3 "Side view" of the NPCs with Nup96, Nup62 and Pom121 imaged by splitSMLM. a** Scheme of the NPCs at the NE. ONM, outer nuclear membrane; INM, inner nuclear membrane. **b** Averaged "side view" profiles of Nup62, Nup96 and Pom121 obtained by alignment of 65 individual NPCs. **c** U2OS cell with immuno-fluorescently labeled Pom121 (blue), Nup62 (red) and Nup96 (green) with zoomed-in "side view" regions in insets. Scale bars, 1 µm (**c**) and 100 nm (insets).

graphical user interface and can be used as a stand-alone application for Windows, or can be run under Matlab or from SharpViSu[23]. See Supplementary Methods for download and installation instructions.

The software interface guides the user through all the steps of demixing, starting from visualization of input data and flexible selection of the input regions for each channel of the image splitter. The two input datasets can then be automatically or manually aligned and the localizations can be paired within a user-defined distance tolerance. The resulting paired localizations are visualized in the form of a bivariate histogram of photon counts, using either linear or logarithmic scales of axis and colormaps. The fluorophores are assigned using user-defined flexible sector regions on the bivariate histogram, and those sectors can be saved and reused for further experiments. SplitViSu estimates the cross-talk between demixed channels if

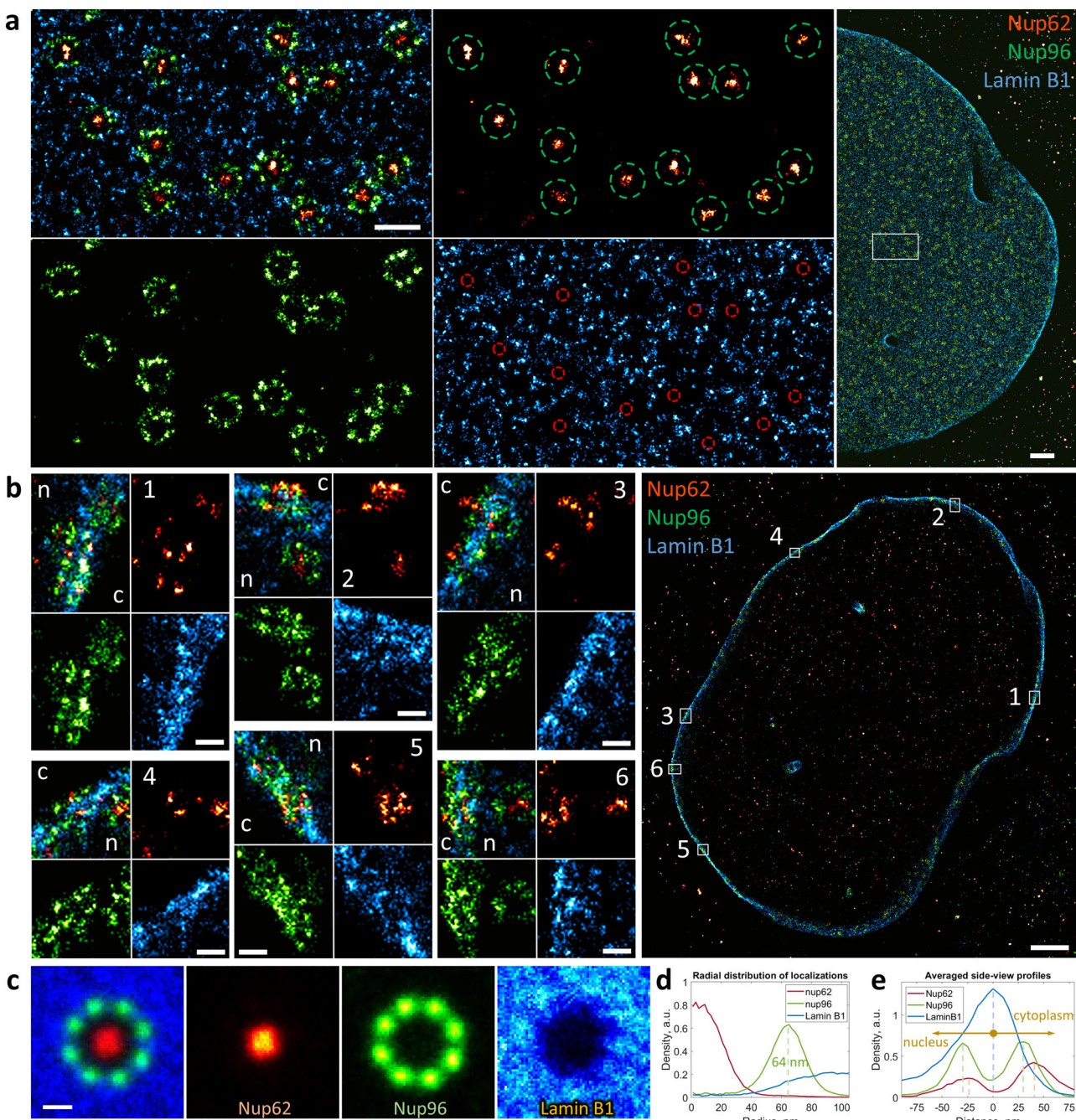

**Fig. 4 Three-color splitSMLM imaging of lamin B1 at the NE. a** "Top view" of the NE (Nup96, green; Nup62, red; lamin B1, blue). Green circles with a diameter of 150 nm represent the positions of the Nup96 ring. Red circles have a diameter of 50 nm and represent the positions of Nup62. White rectangle indicates the zoomed-in region. **b** "Side views" of the NE. The numbered rectangles in the general view represent the zoomed-in regions; n, nucleus; c, cytoplasm. **c** Sum of aligned images of individual NPCs. **d** Radial distribution of localizations of each protein obtained by averaging 99 NPC particles. **e** Averaged "side view" profiles of Nup62, Nup96 and lamin B1 obtained by alignment of 66 individual NPCs. Scale bars, 200 nm (**a**, left), 1 μm (**a**, right and **b**, right), 100 nm (**b**, left), 50 nm (**c**).

data from a single-labeled sample is used together with sector regions for 2–3 output channels (Supplementary Fig. 6). SplitViSu allows to choose a method for calculation of output coordinates, e.g. using the coordinates from only one input channel or using their simple or weighted mean positions. For the "weighted mean" method, a residual chromatic error can be automatically detected and removed (Supplementary Fig. 4). Finally, the demixed localizations can be saved on disk or exported to SharpViSu[23] for further processing.

**Spectral properties of filters for image-splitting optics.** The emission spectra of AF647, CF660C and CF680, like of many fluorophores, have a rapid increase that starts near their excitation maximum, followed by a longer tail after the emission peak (Supplementary Fig. 1b). In order to efficiently discriminate their spectra, it is important to detect the spectral region where they differ most, i.e., the region between their excitation and emission maxima. At the same time, this region lies closely to the excitation laser line and is usually rejected by emission filters

(Supplementary Fig. 11). This not only rejects precious emitted photons but also makes the spectral discrimination less reliable. Detection of emission within this spectral region ("salvaged fluorescence") has been shown to be beneficial for spectral separation but up to now required a rather complex optical setup[21].

In this work, we managed to detect the "salvaged fluorescence" region by carefully selecting filters on a conventional epifluorescence setup that allowed to detect fluorescence starting from 652 nm, only 10 nm apart from the excitation laser and to better discriminate fluorophores with very close emission spectra (Supplementary Figs. 11, 12). To separate the emission light into two channels, we used a dichroic mirror at 690 nm that provides, for the used fluorophores, similar intensities and background in both channels (Fig. 1f, Supplementary Figs. 1b and 11). This allows using a single run of a conventional fitting algorithm for both channels at the same time. For spectral detection close to a laser line, conventional filters usually do not provide low enough background for single-molecule observation. Each combination of filters must be carefully designed and tested. In our setup, we therefore used an additional band-pass filter in the $\lambda_S$ channel of the image splitter in order to remove residual background from the laser (Supplementary Fig. 1a).

**Removal of unreliable localizations by spectral detection**. Interestingly, detection of individual fluorophores through a spectral image splitter is beneficial even for single-labeled samples (Fig. 1b). Firstly, because a given fluorophore species has a unique ratio $r = I_L / I_S$ (1), the localizations with a different ratio might originate from a different species of dyes (e.g. from fluorophores that cause autofluorescence) or from spurious localizations (Supplementary Fig. 13). Secondly, in our setup, we introduced a condition that, in order to be considered for the analysis, a molecule must be detected in both channels independently. A small tolerance (50–100 nm) between the channels is necessary to account for small chromatic errors and for the localization imprecision in the input data. This provides a strong filter to reduce artefactual localizations originating from background noise or from poorly blinking regions (Fig. 1b, Supplementary Fig. 13). The localizations originating from background noise can be detected at random places, therefore they are unlikely to be detected at the same place in both channels at the same frame, which provides a way to exclude them. Poorly blinking regions, depending on the localization algorithm, can cause erroneous localizations at different positions within two channels; they might not fall within the tolerance and would be removed as well. Also, multiple blinkings involving more than one fluorophore species usually have a different $r$ value and can also be filtered out. It should be noted, however, that for samples with 2–3 and more fluorophores the removal of unreliable localizations becomes less efficient because the "unreliable" region for one fluorophore may coincide with the "reliable" region for the other dye; yet, part of the autofluorescence is removed in between the selected sectors within the histograms, which improves data quality.

**Optimized buffer composition for simultaneous multi-color imaging**. Sample preparation is a key factor because imaging buffers play a huge role in the behavior of fluorophores in SMLM[36–38] with rather variable performances for different dyes[37–39]. Since in multi-color SMLM the performance of fluorophores affects not only their localization but also their spectral separation, it is important to choose buffers to maximize the photon yield of the fluorophores and yet obtain comparable photodynamics between different fluorophores.

While the classical Glox-MEA buffer provides already acceptable performance for image splitter-based SMLM[22], the presence of slowly-blinking low-intensity localizations of CF680 and CF660C[22] is problematic because those molecules cannot be reliably demixed and should be rejected. Moreover, we find that AF647, CF660C and CF680 in this buffer demonstrate different reactivation responses on illumination with 405 nm light: AF647 can be reactivated continuously with gradual increase of the reactivation intensity (e.g., in the 0 to 50 mW range, see methods), while CF660C and especially CF680 are reactivated well with very low power of the laser, but cannot be reactivated anymore when the power increases to intermediate or high values (Supplementary Fig. 14). In order to keep the density of localizations per frame nearly constant, the reactivation power is usually increased gradually during an acquisition. However, the non-equivalent response of fluorophores on reactivation leads to a non-constant ratio between the densities of the dyes during the experiment, i.e., there are more CF680 molecules at the beginning of an acquisition while the AF647 molecules are much more frequent at the end. This might lead to undesirable effects, for example, when the drift is not perfectly corrected or when, to save time, the acquisition has been stopped before complete bleaching of all three fluorophores.

In order to improve the performance of the fluorophores, we tested a variety of conditions and found that addition of 2 mM of cyclooctatetraene (COT) to the Glox buffer increases the photon yield of all three fluorophores by 20–50% without decreasing the density of localizations (Supplementary Fig. 14a, c), an effect previously observed only for AF647[36]. Moreover, we show that COT equalizes the response of these fluorophores on reactivating light. In Glox with COT, AF647 slightly decreases its response at high 405 nm intensities, while CF660C and CF680 substantially increase it (Supplementary Fig. 14b, c). COT increases the proportion of bright molecules for all three fluorophores (Supplementary Fig. 14) and allows for more reliable demixing and reduces rejection of localizations (Supplementary Fig. 15). Importantly, we found that COT does not produce any additional background in SMLM (Supplementary Fig. 16).

**Application of optimized three-color imaging to the nuclear pore complex**. We applied our method for simultaneous three-color imaging of individual proteins within the nuclear envelope (NE) of human cells. Thanks to the ellipsoidal shape of the NE, one can observe the NPCs projected under different angles: "top views" when focusing on the NE close to the coverslip and "side views" when focusing on the middle of the nucleus, several micrometers away from the coverslip (see scheme, Fig. 2b, 3a). NPCs have dimensions of ~150 nm and a partially symmetrical structure that makes it a perfect test object for super-resolution microscopy[24]. Within the NPC, a number of different proteins can be detected, providing a range of distances that can be used as nanometer-scale rulers for various imaging and labeling tests. We chose to label NPC subunits that correspond to distinct parts of NPC such as Nup96, Nup62 and Pom121 (Fig. 2b, 3a) as well as lamin B1, a protein found close to the inner nuclear membrane.

Our SMLM images of Nup96 demonstrate 8-fold rotational symmetry with respect to the central axis of the NPC and two-fold symmetric localization with respect to the NE (Figs. 2–4), in line with the structure of the NPC (Fig. 2b) and with previous super-resolution studies[24]. Nup62 appears in the center of the NPC in the "top view" of our super-resolution images (Figs. 2, 4a, c, d), as can be expected for a central channel protein of the NPC. In the "side view", however, it can be seen at both the

nucleoplasmic (weaker signal) and the cytoplasmic sides (stronger signal) of the complex (Figs. 3 and 4b, e), indicating the presence of its copies at both the nuclear and the cytoplasmic sides of the NPC.

Next, our data suggests that Pom121 is located only at the nuclear side of the NE, close to the nuclear ring of Nup96 (Fig. 3b, e), confirming the role of Pom121 as an anchoring point for the NPC at the inner nuclear membrane[40,41]. In the "top view", Pom121 forms an irregular ring structure between the regular Nup96 ring and the NPC central channel, revealed by the Nup62 staining (Fig. 2d–f). The Pom121 localization inside the NPC in this projection corresponds to the deep protrusion of the NE into the NPC between the cytoplasmic and the nuclear rings of the Y-complexes[42] (Fig. 2b). From the "top view" of our three-color images it is evident that lamin B1 is abundant at the NE, but it is completely absent precisely where NPCs are located (Fig. 4a, c, d), which is in line with the current model of the inner nuclear membrane region (Fig. 3a). In the "side view", however, we find that lamin B1 can be found in the center between the nuclear and the cytoplasmic rings of the NPC (Fig. 4b, e), closer to the cytoplasm than thought according to current models[43].

## Discussion

Simultaneous multi-color SMLM with spectral image splitting was demonstrated in the early days of super-resolution microscopy[25,26], but it did not spread much beyond experimental setups. In this work, we describe splitSMLM, an image-splitting method for multi-color SMLM that comprises a spectral image splitter, a dedicated software for spectral demixing (SplitViSu), a selected fluorophore triplet with three close wavelengths and an optimized buffer system that keeps all 3 fluorophores active in an equivalent manner. Thanks to simultaneous imaging and improved signal-to-noise ratios because essentially all photons are used, splitSMLM preserves the localization precision and makes drift correction much easier, which is crucial for accurate colocalization studies. The splitSMLM implementation is simple and robust and results in much improved image resolution, drift correction, compensation of chromatic errors at the nanometer scale and performance of fluorophores in optimized buffers.

The setup is built with commercially available components and is therefore easy to implement at biology laboratories. Previous implementations had to make compromises in terms of photon budget, using only a subset of detectable light for localization and therefore losing precision. Moreover, the low brightness and suboptimal photoswitching of previously employed fluorophores led to under-detection and strong rejection of localization events, resulting in labeling density loss. The splitSMLM method presented here overcomes many of these limitations. It allows convenient simultaneous rather than sequential multi-color SMLM acquisitions without the need of separate acquisitions for each fluorophore at longer and shorter excitation and emission wavelengths, respectively[13,44,45]. The localization density is preserved and precision is improved as visible in the images unlike in previous methods using reduction, which lead to intensity loss (Supplementary Fig. 8). Therefore, the good localization density and precision provide better resolution. The method reduces the acquisition time up to three times as compared to sequential three-color imaging, which has an important practical effect because high-quality SMLM requires long acquisitions (minutes to hours)[22]. The fact that the drift is the same in all channels allows summing all channels for drift estimation resulting in higher precision for each channel. Moreover, the present approach uses a specifically combined spectrally distinct

fluorophore triplet (AF647, CF660C, CF680) for which conjugated antibodies are available and which achieves equally high performances thanks to an optimized buffer that is identical for the entire experiment. The proposed imaging with three fluorophores addresses the majority of cases needed in practice for immunofluorescence experiments (Figs. 2–4).

An important point to consider in the splitSMLM method is the combination of optimized choice of the fluorophore triplet and optimally adapted emission filters, which allows detecting fluorescence very close to the laser line (close to 10 nm), thereby achieving "salvaged fluorescence" with maximized detection and improved spectral separation of the fluorophores. The dichroic image-splitting mirror was chosen to split the emission into two channels with similar spectral widths, allowing reliable detection of molecules within both channels simultaneously (Supplementary Figs. 11 and 12). Furthermore, the optimized imaging buffer that comprises an addition of COT into the Glox-MEA buffer increases the photon yield of all three fluorophores and equalizes the response of fluorophores on reactivation, thus allowing colocalization analysis under rather stable imaging conditions. The optimized imaging buffer composition hence improves the photon budget and the amount of reactivation of far-red fluorophores, which is beneficial for splitSMLM. COT therefore has an enhancing photophysical effect (Supplementary Figs. 14, 15).

To analyze splitSMLM data conveniently, we developed a new demixing software, SplitViSu. This tool allows visualization of input data, flexible selection of the regions for each channel, automatic or manual alignment of localizations between the channels, pairing of localizations within a given distance, visualization of the paired localizations as a bivariate histogram of intensities, flexible assignment of the localizations to fluorophores based on the photon counts and their ratios, and estimation of (unwanted) cross-talk between demixed localizations (Supplementary Fig. 10). The software can be used as a stand-alone application under Windows, or can be run under Matlab or from the SharpViSu software[23] in which it was added as a plugin. The demixing algorithm uses all detectable photons both for demixing and localization, improves the localization precision and corrects chromatic errors. The usage of similar wavelengths has the specific advantage here of reducing lateral chromatic aberration thanks to the capability of the software to demix spectrally close fluorophores, which is improved even further by additional processing in SplitViSu. Moreover, we introduced the refinement of re-localization events, which further increases image resolution (Fig. 2e). Additionally, we show the benefits of the image-splitting setup and fluorophore demixing also for single-labeled samples (Fig. 1b and Supplementary Fig. 13) thanks to filtering out localizations that originate from background noise or from autofluorescence. Unlike other software with some similar functionalities[18,20], the SplitViSu tool provides flexibility in calculation of output localizations, with the possibility of freely selecting options to use only coordinates from a selected channel or use their mean or weighted mean values, thus improving the localization precision and compensating chromatic errors. Moreover, SplitViSu is fully integrated with the SharpViSu[23] workflow, allowing drift correction, visualization, estimation of resolution, filtering, segmentation & cluster analysis with ClusterViSu and 3DClusterViSu[12,46] and further processing of spectrally demixed multi-color data. The SplitViSu software is made freely available under https://github.com/andronovl/SharpViSu; see Supplementary Methods for download and installation instructions.

The strength of the splitSMLM method described here relies on a strong synergistic effect of several parameter that is obtained from the much-improved multi-color super-resolution data quality, achieved by refining each aspect of multi-color imaging—

optics, data processing and performance of fluorophores. The quality improvements include (1) higher image resolution; (2) higher density of localization; (3) minimal cross-talk between the labels; (4) no visible chromatic distortion; (5) suppression of localizations originating from background noise or from auto-fluorescence. The limitations of splitSMLM include: (1) potential complication of the detection path of the microscope that however can be overcome with commercial image splitters; (2) loss of some minor localizations during demixing (Supplementary Figs. 6, 7); (3) need for additional data processing steps that are, however, now implemented in the free software SplitViSu; (4) a choice of specific fluorophores, but these may be extended in the future.

When this paper was under revision, another work was published that uses a similar method for computation of the output localizations using the weighted mean of the input localization pairs with automatic correction of chromatic errors[47]. The procedure was applied to triple-color DNA-PAINT and double-color dSTORM data. The two methods, splitSMLM and SD-DNA-PAINT[47], are therefore the first to our knowledge multi-color demixing methods that use all available photons for localization and offer automatic correction of chromatic distortions. In this work, we provide a dedicated software, SplitViSu, that integrates all the demixing processing steps in a single interactive tool to facilitate the workflow and that can be used for any SMLM experiment type (e.g., for STORM, PALM, PAINT or MIN-FLUX). Furthermore, we extend spectral demixing-based dSTORM to three colors using the CF660C fluorophore; we improve the performance of all fluorophores using a COT-supplemented imaging buffer (in our hands, in this buffer CF680 can be replaced also by Alexa Fluor 680 with similar performance) and appropriate dichroic filters and we increase the resolution of demixed data thanks to refinement of multiple localizations.

We applied our new approach to three-color super-resolution imaging of the NPC and the nuclear lamina. The obtained splitSMLM data validate our experimental approach, confirming as expected[24] the typical 8-fold localization of Nup96 in the "top view" and its symmetrical localization with respect to the NE in the "side view" (Figs. 2–4). Furthermore, the central localization of Nup62 within the NPC (Figs. 2–4) and absence of lamin B1 at the nuclear pores in the "top view" (Fig. 4a, c, d) now achieve a refined positioning of the individual proteins within the nuclear membrane as compared to earlier data[48]. Three-color imaging and colocalization of Pom121 has not been done previously (only with single-color microscopy[49,50]) and now reveals the presence of irregular ring structures between the Nup96 ring and the central channel, represented by Nup62 (Fig. 2d–f). In the "side view", it is visible that Pom121 localizes only at the nuclear side of the NPC (Fig. 3b, c), which suggests that Pom121 clusters act as NPC deposition loci at the inner nuclear membrane, providing unprecedented insights not seen in previous super-resolution studies. Additionally, some Pom121 structures are resolved at the NE consistent with the composition of normal NPCs, but with fewer localizations of Nup96 and Nup62 (Fig. 2c, d, circles). Considering that Pom121 acts as a deposition site of the NPC, these structures might correspond to the NPCs observed at an early stage of their deposition to the NE, providing first insights into the NPC assembly pathway. Regarding lamin B1, which is a type V intermediate filament in the nuclear lamina that lines the inner surface of the NE relevant for chromatin organization[51], previous SMLM studies showed that it lays close to the nuclear membrane but its localization with respect to the NPC remained unclear[52]. Here we show with precise three-color imaging that it localizes very closely to the nuclear membrane and in the center

between the cytoplasmic and the nuclear rings of Nup96 as visible in the "side views" (Fig. 4b, e).

Taken together, the analysis of NPC components Nup96, Nup62, Pom121 and lamin B1 illustrates the strength of our spectral demixing method for high-precision multi-color localization microscopy of challenging, multi-component cellular complexes. The splitSMLM method will be widely applicable, e.g. for a multitude of colocalization studies requiring multi-color SMLM and super-resolution imaging in biology.

## Methods

**Maximizing the localization precision and minimizing chromatic errors in multi-color imaging**. Let us regard each single-molecule localization as a normally distributed random variable

$$\mathbf{X} = N(\boldsymbol{\mu}, \sigma^2) \tag{5}$$

where mean $\boldsymbol{\mu}$ is the position of the fluorophore and the standard deviation

$$\sigma = \sigma_0 / \sqrt{I} \tag{6}$$

where $\sigma_0$ is the standard deviation of the point spread function and $I$ is the number of detected photons. With a dichroic image splitter, for each molecule, there are two experimental localizations,

$$\mathbf{X_L} = N(\boldsymbol{\mu_L}, \sigma_L{}^2) \tag{7}$$

and

$$\mathbf{X_S} = N(\boldsymbol{\mu_S}, \sigma_S{}^2) \tag{8}$$

that originate from the $\lambda_L$ and the $\lambda_S$ channels, respectively. In an ideal image splitter without losses, all input photons are split into two channels with a dichroic mirror:

$$I = I_L + I_S \tag{9}$$

The ratio $r = I_L / I_S$ (1) depends on the spectral characteristics of the mirror and on the emission spectra of the fluorophore. The two positions of a fluorophore imaged through the splitter do not coincide due to chromatic errors (which comprise lateral chromatic aberrations and mirror imperfections): $\boldsymbol{\mu_S} \neq \boldsymbol{\mu_L}$ (Figs. 1f, 2a and Supplementary Fig. 5). The demixing algorithm decides, based on $I_L$ and $I_S$, to which species the localization belongs, and then calculates the position of the demixed localization based on its $\boldsymbol{X_L}$ and $\boldsymbol{X_S}$. We will neglect here the chromatic aberrations between fluorophores acquired through the same channel, because the average wavelength is very close for different fluorophores within one channel (Fig. 1f and Supplementary Fig. 3). We will only consider shot noise as it cannot be avoided. We will neglect other noise and background sources since those are not fundamental and can be made minimal for a given setup and sample.

In the case where all demixed localizations adopt coordinates from the $\lambda_L$ channel,

$$\mathbf{X} = \mathbf{X_L} = N(\boldsymbol{\mu_L}, \sigma_L{}^2) \tag{10}$$

there is no chromatic error ($\boldsymbol{\mu} = \boldsymbol{\mu_L}$), but the localization precision is decreased:

$$\sigma = \sigma_0 / \sqrt{I_L} = \sigma_0 / \sqrt{(I - I_S)} > \sigma_0 / \sqrt{I} \tag{11}$$

A similar result is obtained when all localizations adopt coordinates from the $\lambda_S$ channel (Fig. 2a). To have best localization precision ($<\sqrt{2}$ times worse than when all photons are used for localization), one would choose for localization the channel that has higher intensity. In many cases, however, the channel with highest intensity for one fluorophore will be the channel with lowest intensity for another fluorophore. Now, if one chooses to localize this second fluorophore within the channel that has highest intensity, the chromatic error will come into play because different species will be localized from spectrally different channels. One would often rather "sacrifice" the localization precision in order to avoid chromatic shift and would lose $>\sqrt{2}$ times in localization precision for the fluorophores that have less intensity in the channel used for localization[16,32].

In the following, we describe a way to utilize the photons present in both channels for both spectral demixing and localization. We developed an algorithm to calculate the coordinates of the demixed localization as a weighted sum of its coordinates in the two spectral channels:

$$\mathbf{X} = w_S \cdot \mathbf{X_S} + w_L \cdot \mathbf{X_L} \tag{12}$$

where $w_S$ and $w_L$ are intensity-dependent weights that can adopt values from 0 to 1:

$$w_S = I_S / (I_S + I_L), w_L = I_L / (I_S + I_L); w_S + w_L = 1 \tag{13}$$

The resulting localizations are normally distributed

$$\mathbf{X} = N(w_L \cdot \boldsymbol{\mu_L} + w_S \cdot \boldsymbol{\mu_S}, w_L{}^2 \cdot \sigma_L{}^2 + w_S{}^2 \cdot \sigma_S{}^2) \tag{14}$$

and are localized in between of the two initial localizations, thus resulting in reduced chromatic error

$$\mu = w_L \cdot \mu_L + w_S \cdot \mu_S \tag{15}$$

The chromaticity is not completely removed though, because the position of the demixed molecules becomes dependent on the ratio of their intensities in the $\lambda_S$ and $\lambda_L$ channels:

$$\mathbf{X} = \mathbf{X_S}/(1 + r) + r \cdot \mathbf{X_L}/(1 + r) \qquad (16)$$

where $r = I_L / I_S$ (1). It is corrected better for fluorophores that have closer $r$-values (fluorophores with closer emission spectra).

Importantly, the localization precision in this (ideal) case approaches that for imaging without image splitter, the difference being only due to intensity losses in the image-splitting optics that can be made minimal (Supplementary Fig. 1b):

$$\sigma = \sqrt{w_L^2 \frac{\sigma_0^2}{I_L} + w_S^2 \frac{\sigma_0^2}{I_S}} = \sqrt{I_L \frac{\sigma_0^2}{(I_L + I_S)^2} + I_S \frac{\sigma_0^2}{(I_L + I_S)^2}} = \frac{\sigma_0}{I_L + I_S} \sqrt{I_L + I_S}$$
$$= \frac{\sigma_0}{\sqrt{I_L + I_S}} = \frac{\sigma_0}{\sqrt{I}} \qquad (17)$$

Therefore, the weighted mean method fully preserves the localization precision and yet reduces chromatic shifts (Fig. 2a).

Now let us consider a case where the demixed coordinates are calculated as a simple mean of $X_S$ and $X_L$:

$$\mathbf{X} = (\mathbf{X_S} + \mathbf{X_L})/2 = N\big((\boldsymbol{\mu_L} + \boldsymbol{\mu_S})/2, (\sigma_L{}^2 + \sigma_S{}^2)/4\big) \qquad (18)$$

The chromatic error would be completely corrected for all localizations—the demixed localization is located in the middle between the two input localizations:

$$\mu = (\mu_L + \mu_S)/2 \qquad (19)$$

The localization precision, however, would be decreased, unless $I_L = I_S$:

$$\sigma = \frac{\sigma_0}{2} \sqrt{\frac{1}{I_L} + \frac{1}{I_S}} \geq \frac{\sigma_0}{\sqrt{I}} \qquad (20)$$

In practice, it is important to know under which conditions the localization precision using the 'mean' method will be higher than when using the coordinates of only the brightest channel. Suppose $I_L > I_S$, then the condition for the improvement of the localization precision can be written as:

$$\frac{\sigma_0}{2} \sqrt{\frac{1}{I_L} + \frac{1}{I_S}} < \frac{\sigma_0}{\sqrt{I_L}} \Rightarrow \frac{1}{2} \sqrt{\frac{I_L}{I_L} + \frac{I_L}{I_S}} < 1 \Rightarrow \sqrt{1 + r} < 2 \Rightarrow 1 + r < 4 \Rightarrow r < 3 \qquad (21)$$

The localization precision using the 'mean' coordinates will be improved if the ratio of the intensity between the $\lambda_S$ and $\lambda_L$ channels for the given fluorophore is <3, which is the vast majority of cases in practice. It should be noted that using this method, the localization precision is always improved by a factor > $\sqrt{2}$ if a fluorophore has most of its emission in the channel that is not used for localization but only used for demixing in the previously reported demixing algorithms.

The coordinates calculated by the "weighted mean" method exhibit residual chromatic error equal to the lateral chromatic aberration between the fluorophores imaged without image splitter (Supplementary Fig. 3), plus an unequal magnification of the channels that might be produced by the image-splitting optics. For a given fluorophore with a constant ratio $r$, the residual shift depends only on the fluorophore's position within the image and can therefore be registered using information already available in the data. We register the shift with respect to the position, calculated using the "simple mean" method, as it provides the position closest to the ground truth for all fluorophores (Fig. 2a, Supplementary Fig. 3). For every fluorophore within a dataset, we calculate their simple and weighted mean positions, and the difference of those positions:

$$\Delta \mathbf{x} = \mathbf{x_{wm}} - \mathbf{x_m} \qquad (22)$$

$\Delta \mathbf{x}$ has a random component due to the limited localization precision and due to fluctuations in the photon counts, and a non-random component that depends on the position $x$ due to the chromatic error:

$$\Delta \mathbf{x} = \Delta \mathbf{x_{rand}} + \Delta \mathbf{x_{chro}} \qquad (23)$$

The typical values for our setup and fluorophore triplet are:

$$|\Delta x_{rand}| \leq 20 nm \qquad (24)$$

$$|\Delta x_{chro}| \leq 4 nm \qquad (25)$$

(Supplementary Fig. 4). We fit the chromatic component $\Delta \mathbf{x_{chro}}$ as a function of the coordinates with a polynomial, separately for $x$ and $y$ coordinates, and subtract it from the "weighted mean" positions:

$$\mathbf{X} = \mathbf{x_{wm}} - \Delta \mathbf{x_{chro}} \qquad (26)$$

(Supplementary Fig. 4).

**Cell culture and immunofluorescence**. U2OS-CRISPR-NUP96-mEGFP cells[24] (300174, CLS) were plated in glass-bottom petri dishes (Cellvis D35-14-1.5-N). At ~50% confluency the cells were washed twice in phosphate-buffered saline solution (PBS), fixed with 1% formaldehyde in PBS for 15 min and washed three times in PBS. The cells were permeabilized with 0.1% Triton X-100 in PBS (PBS/Tx) for 15 min and then blocked with 3% bovine serum albumin (BSA) in PBS/Tx (PBBx) for 1 h. The primary antibodies were incubated, unless otherwise stated, at 2 µg/ml

in PBBx overnight at +4 °C. The samples were washed three times with PBBx for at least 5 min each time. Then the secondary antibodies were incubated at 4 µg/ml in PBBx for 2 h and the cells were washed again three times with PBS/Tx for at least 5 min each time. The samples were then postfixed with 1% formaldehyde in PBS for 10 min, washed twice with PBS and kept in PBS at +4 °C until mounting for imaging.

The following antibodies were used: Mouse anti-Nup62 (610497, BD Biosciences); Rabbit anti-GFP (TP-401, Torrey Pines Biolabs); Chicken anti-GFP (GFP-1010, Aves Labs); Mouse anti-GFP (2A3, IGBMC); Rabbit anti-lamin B1 (12987-1-AP, Proteintech), used at 0.7 µg/ml; Rabbit anti-Pom121 (GTX102128, GeneTex); Goat anti-Mouse CF680-conjugated (SAB4600199, Sigma); Goat anti-Rabbit CF680-conjugated (SAB4600200, Sigma); Goat anti-Rabbit CF660C-conjugated (SAB4600454, Sigma); Goat anti-Mouse CF660C-conjugated (SAB4600451, Sigma); Goat anti-Chicken CF660C-conjugated (SAB4600458, Sigma); Goat anti-Rabbit AF647-conjugated (A-21245, Thermo Fisher); Goat anti-Mouse AF647-conjugated (A-21236, Thermo Fisher).

**Imaging buffers**. The samples were imaged in a water-based buffer that contained 200 U/ml glucose oxidase, 1000 U/ml catalase, 10% w/v glucose, 200 mM Tris-HCl pH 8.0, 10 mM NaCl, 50 mM MEA with addition of 2 mM COT. The mixture of 4 kU/ml glucose oxidase (G2133, Sigma) and 20 kU/ml catalase (C1345, Sigma) was stored at −20 °C in an aqueous buffer containing 25 mM KCl, 4 mM TCEP, 50% v/v glycerol and 22 mM Tris-HCl pH 7.0. MEA-HCl (30080, Sigma) was stored at a concentration of 1 M in $H_2O$ at −20 °C. COT (138924, Sigma) was stored at 200 mM in dimethyl sulfoxide at −20 °C.

The samples were mounted immediately prior imaging using ~200 µl of the imaging buffer and placing a clean coverslip on top of it while avoiding bubbles. The interface between the plastic bottom of the petri dish and the clean coverslip in a well with imaging buffer is sufficiently airtight thanks to the surface tension and allows for long imaging runs (≥8 h) without pH shift. After imaging, the samples were rinsed once with PBS and kept in PBS at +4 °C.

**Super-resolution imaging**. The SMLM imaging was performed on a modified Leica SR GSD system (Supplementary Fig. 1). We used an HCX PL APO 160x/1.43 Oil CORR TIRF PIFOC objective that provides an equivalent pixel size of 100 nm on the camera. The fluorescence was excited with a 642 nm 500 mW fiber laser (MBP Communication Inc.) and reactivated with a 405 nm 50 mW diode laser (Coherent Inc.). We used a custom double-band filter cube (ordered and assembled at AHF Analysentechnik AG) with a dichroic mirror (DM1) Semrock FF545/650-Di01 and emission filters (F1) Semrock BLP01-532R mounted (airspaced) together with Chroma ZET635NF. The fluorescence was split into two channels with an Optosplit II (Cairn Research) image splitter, attached to a camera port of a Leica DMI6000B microscope. After splitting with a Chroma T690LPXXR dichroic mirror (DM2) and filtering the $\lambda_S$ channel with a Chroma ET685/70m band-pass filter (F2), both channels were projected side-by-side onto an Andor iXon Ultra 897 (DU-897U-CS0-#BV) EMCCD camera. For measurements without image splitter, after the filter cube, the fluorescence was additionally filtered with a Semrock BLP01-635R-25 long-pass filter (F3) and the images were recorded with an Andor iXon+ (DU-897D-C00-#BV) EMCCD camera, attached to the second camera port of the microscope.

For SMLM imaging, the sample was illuminated with a constant power (30–50%) of the 642 nm laser. The first seconds were not recorded to avoid the initial high density of localizations. The acquisitions were started manually after appearance of separate single-fluorophore events ("blinking"). The exposure time of the camera was 8–13 ms; the electron multiplying gain of the camera was 300x (iXon Ultra) or 100x (iXon+). After 10–20 min, as the number of localizations dropped, the sample started to be illuminated additionally with a 405 nm laser with gradual increase of its intensity in order to keep a nearly constant rate of single-molecular returns into the ground state. The acquisition was stopped after almost complete bleaching of the fluorophore. Slight axial drifts were compensated by manual re-focusing, keeping in focus the localizations originating from the bottom nuclear envelope.

For comparison between the imaging buffers, exactly the same parameters were kept for all acquisitions: the Andor iXon+ camera, EM gain 100x, exposure time 12.117 ms, 642 nm laser at 40% power. The first $10^5$ frames were acquired only with the 642 nm laser excitation, without a 405 nm reactivation. For the photoinduced reactivation experiments (Supplementary Fig. 14b), the same region was acquired a second time after the first acquisition. The intensity of the 405 nm laser was set to 0% at the beginning and then increased to 4% at about frame #1000, to 4.5% at frame #13000, 5% at #20000, 6% at #30000, 7% at #40000, 8% at #50000, 10% at #60000, 15% at #70000, 20% at #75000, 30% at #80000, 50% at #85000, 75% at #90000, 100% at #95000. The power of the 405 nm laser in % was set in the LAS X wizard, it does not correspond linearly to the actual laser power, but provides the same power at same values, that is sufficient for comparison between different imaging buffers.

**Data processing**. The fitting of single-molecule images was performed in the Leica LAS X software with the "direct fit" fitting method, using an appropriate detection threshold. The localization tables were then exported for further processing in the

SharpViSu software workflow[23] and with customized Matlab procedures as implemented in SplitViSu. If imaged with the image splitter, the localizations were first demixed with the SplitViSu plugin and then processed in the main SharpViSu software for drift correction and for re-localizations on consecutive frames. Since the drift is the same for all demixed channels, for drift correction in SharpViSu, we added possibilities to calculate the drift based on data in one of the channels or on the sum of data in all demixed channels. The calculated by cross-correlation drift is then subtracted from localizations of all channel. For the data in this paper, we calculated the drift based on the sum of localizations in all channels.

For comparison between the imaging buffers (Supplementary Figs. 14 and 15), all localizations were detected using a threshold of 25 photons/pixel in the LAS X wizard. After drift correction, the datasets were cropped to include only the surface of a nucleus with labeled nuclear pores (one nucleus per dataset). The photon counts and other localization statistics were calculated within these cropped regions, and the occurrence of localizations were normalized on the area of the corresponding regions, in order to get densities comparable between various datasets. All detected localizations within the region were assessed, without correction for re-localizations in consecutive frames and without any filtering. For all other datasets, we used a search radius of 50 nm for finding the re-localizations in the consecutive frames with their subsequent refinement. The resulting super-resolution images were built as 2D histograms of the single-molecule coordinates.

The emission spectra of fluorophores, observed in the $\lambda_S$ or the $\lambda_L$ channels, were calculated as a multiplication of the normalized emission spectrum of the fluorophores with the transmission of the dichroic mirrors and filters used in the given channel and the quantum yield of the camera. The average emission wavelength in a given spectral channel was calculated as

$$\lambda_0 = \int_{630nm}^{800nm} \lambda \cdot I(\lambda) d\lambda / \int_{630nm}^{800nm} I(\lambda) d\lambda \qquad (27)$$

where $\lambda$ is the wavelength and $I(\lambda)$ is the emission of the fluorophore observed in the given channel. For the case without image splitter, $I(\lambda)$ was calculated considering the emission spectra of the fluorophores, transmission spectra of DM1 and F1 and the quantum yield of the camera.

For averaging of multiple NPCs in the "top view", the individual NPCs were picked manually, keeping localizations within a radius of 130 nm from the manually picked center of each NPC. The obtained Nup96 particles were aligned with smlm_datafusion2d[53] imposing 8-fold symmetry (random rotation of every particle by $n$ 45°, $n = [0, 7]$, after each alignment iteration). The localizations of co-imaged proteins were transformed using these alignment parameters of Nup96. The radial distribution of localizations was calculated as a histogram of the radius of the particles in polar coordinates. In order to obtain values proportional to the image profile, every histogram bin was additionally divided by the surface area of the corresponding concentric annulus. For averaging of multiple NPCs in the "side view", the profiles of individual particles along the NPC's axis were calculated in Fiji[54], averaging through the whole diameter of the NPC. The profiles of Nup96 images were fitted with a sum of two Gaussians in Matlab. The center between the Gaussians was set as the new origin of the x-axis and all particles were summed after this alignment. The profiles of co-imaged proteins were transformed using these alignment parameters of Nup96 and were summed as well.

**Statistics and reproducibility**. Where applicable, the final data are shown as mean ± standard deviation of repeated measurements. The number of measurements is reported in the corresponding figure legends.

FRC resolution values (Fig. 2e, Supplementary Figs. 5, 8, 9) were calculated in SharpViSu[23] according to the FRC$_{1/7th}$ criterion[33], using the histogram image reconstruction method with a pixel size of 5 nm and 90 frequency steps. The calculation was repeated ten times for every dataset and the resolution was represented as a mean ± standard deviation of the obtained values for the corresponding image. For Supplementary Fig. 9, the mean and standard deviation were calculated from FRC resolution values of five different datasets for each fluorophore.

**Reporting summary**. Further information on research design is available in the Nature Research Reporting Summary linked to this article.

## Data availability
All data are available from the corresponding author on reasonable request. Source data underlying the graphs presented in the main figures are available in Supplementary Data 1.

## Code availability
The latest version of SplitViSu, including the source code, is available under https://github.com/andronovl/SharpViSu. The version of the software described in the paper is available under https://doi.org/10.5281/zenodo.7055709. See Supplementary Methods for download and installation instructions.

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

## Acknowledgements

We thank Yongrong Liao and Izabela Sumara for the anti-Pom121 antibodies; Amelie Zachayus and Arnaud Poterszman for the U2OS-NUP96-mEGFP cell line; Yves Lutz (IGBMC Imaging Platform) for optimization of the labeling protocols and training. This work was supported by the French Infrastructure for Integrated Structural Biology (FRISBI) ANR-10-INSB-05-01 / France 2030 program, the Alsace Region, the infra-structures Instruct-ERIC and iNEXT-Discovery, the CNRS, Association pour la Recherche sur le Cancer (ARC), Institut National du Cancer (INCa), the Fondation pour la Recherche Médicale (FRM), and USIAS of the University of Strasbourg (USIAS-2018-012). This work of the Interdisciplinary Thematic Institute IMCBio, as part of the ITI 2021-2028 program of the University of Strasbourg, CNRS and Inserm, was supported by IdEx Unistra (ANR-10-IDEX-0002), and by SFRI-STRAT'US project (ANR 20-SFRI-0012) and EUR IMCBio (ANR-17-EURE-0023) under the framework of the France 2030 Program.

## Author contributions

L.A. designed research, software and analyzed data. L.A. and D.H. set up the optics. R.G. and L.A. performed fluorescence labeling and imaging. L.A. and B.P.K. wrote the manuscript with input from all authors. B.P.K. supervised the study.

## Competing interests

The authors declare no competing interests.

## Additional information

                                                                                    13