## [Peer Review File · Communications Biology]

Reviewers' comments:

Reviewer #1 (Remarks to the Author):

Dear Authors,

The review is submitted as an attached PDF file.

With best regards

Reviewer #2 (Remarks to the Author):

In the manuscript "A spectral demixing method for high-precision multi-color localization microscopy applied to nuclear pore complexes" from Andronov et al, the authors describe a new method for multiplexing 2-3 colors in super resolution microscopy, utilizing an image splitter-based approach combined with a weighted algorithmic demixing. They apply this approach for three-color imaging of nuclear pore complex (NPC).

The novelty in the approach is the improved multi-color imaging that more efficiently re-distributes the photons in the 2 channels to each fluorophore. Better photon utilization translates in higher precision localization.

There are multiple commercial systems capable of performing Single molecule localization microscopy (SMLM), however multi-color SMLM remains an open challenge outside specialized labs, owing to the spectral bleed through of fluorophores and consequent incorrect assignment of photons in the super-resolution analysis. The same group has published other articles on SMLM. Here the authors focus on analysis of 3-labeled samples, utilizing commercially available image splitters. This can be useful for other researchers in the SMLM field. In contrast to commonly used filter-based acquisition strategies, the authors propose an alternative analysis that exploits the asymmetry of fluorescence emission spectra across two spectrally separate channels to better re-assign photons to the correct fluorophores.

The authors claim that their approach:

1. improved localization precision & drift correction, compensation of chromatic aberrations
2. optimized performance of fluorophores in a specific buffer to equalize their reactivation rates for simultaneous imaging

Both claims are supported by the analysis of experimental data and technical specifications of the detection.

In summary, the paper presents a novel technique and, if the claims stated by the authors are confirmed (see my major comments below), it represents an original and significant technical advance in the field of super-resolution.

However, there are some fundamental points that must be addressed in order to understand the true advantages of the proposed method. The presence of instrument noise and autofluorescence is only partially discussed, and not included in the mathematical considerations. The addition of cyclooctatetraene (COT) is not fully characterized. Disadvantages or limitations of the technique are also not well described. Metrics of performance are not well reported.

Finally, the article writing requires improvement to cater for the broad audience of Nature Communication Biology, writing can certainly be improved. The "overview" figure is not inclusive of the COT claim, which is part of the abstract. This is likely, in my opinion, to complicate understanding of the manuscript for the broad audience.

I ask the authors to address the following concerns.

Major comments

1. Page 10 " Importantly, the localization precision in this case equals that for imaging without image splitter", this statement could be correct in an ideal, lossless optical system. In reality, however, the two camera channels (λ_S , λ_L) both cross multiple optical elements (filters, beamsplitters, mirrors). Each element has a loss (part of which is reported in Supp. Figure 1B). As such, even if small in nature, the beam splitter introduces a loss of signal compared to a system absent of such beam splitter.

a. The authors should address this point in the main text

2. Page 13 "Spectral properties of filters for image splitting optics" The authors should utilize quantitative technical values from filters/bandpass spectra (already present in the manuscript, see Supp. Fig. 1B) to quantify the improvement of signal utilizing their filter combination. This can be achieved by properly utilizing effectively detected area under the spectrum.

3. Page 13 "Very close emission spectra" (Suppl. Fig. S7) and Suppl. Fig. S7 caption "allows better separation". The authors should quantify the improvement (x-fold or percentage) based on angle or other quantitative metrics.

4. Page 14 "detection .. through a spectral image splitter is beneficial even for single-labelled samples", this is a valuable point of the article, as it extends the importance of the technique from a limited subset of users (multi-color SMLM) to the entire community (SMLM). This aspect in particular for identification and removal of incorrect localizations, autofluorescence and possibly background instrumental noise. Autofluorescence removal and incorrect localizations are partially described (and for autofluorescence shown in Supp. Figure 6). However in other figures with $I_{\{L\}}$, $I_{\{S\}}$ plot the same autofluorescence is not as evident and not indicated.

a. Where is autofluorescence in Fig. 1D, Supp. Fig. 7, Supp. Fig. 10? The authors should address this in the manuscript.

b. Some $I_{\{L\}}$, $I_{\{S\}}$ plots have a selection, some other do not. This should be consistent in the manuscript.

c. Some selections do not fully capture all points, example Supp. Fig. 7, Supp. Fig. 10. What happens to the points not selected? How is the selection performed? The authors should address this in the manuscript.

5. The presence of instrumental noise is evident from the spread ($I_{\{L\}}$, $I_{\{S\}}$) of the linear relation in the plots (e.g. Fig. 1D). This noise is not well discussed in the manuscript and should be part of the consideration in selecting/separating fluorophores. Authors should discuss this in the manuscript.

6. The use of cyclooctatetraene (COT) is not well integrated into the manuscript and not fully characterized. The plots in Supplementary Fig. 8 show changes in histograms of intensities but it is unclear if these changes are due to some "enhancing" photophysical effect that "increases the photon yield of all three fluorophores" or if this change is due to intrinsic signal of COT.

a. The authors should characterize the autofluorescence of COT (if any) and present its histogram of intensities, adding to Supp. Fig. 8 to be more convincing of this effect.

7. Page 18 "the new method significantly reduces the acquisition". The authors should quantify this reduction in absolute or relative terms.

Minor comments

1. Figure 1 does not report COT and does not have a clear outline of the proposed method. This figure can be improved.

2. Overall text can be improved. E.g. "COT turns out to equalize the response on reactivating".

Point to point response to the Reviewers' comments:

Reviewer #1

General comments: Within the manuscript titled “A spectral demixing method for high-precision multi-color localization microscopy applied to nuclear pore complexes” the authors Andronov et al. have addressed important technical aspects of multicolor single molecule localization microscopy (SMLM). They have improved currently available methods for multicolor dSTORM that use a dichroic-based splitter to simultaneously image the localization of spectrally overlapping dyes using one laser line. This dichroic-based method was previously implemented by several groups using different dyes and procedures (e.g. Bossi 2008, Testa 2010, Lampe 2012). The success of this method relies on the efficient intensity-based separation of single emitter signals using a process that is currently referred to as ‘spectral demixing’. In general, the manuscript has great potential for publication as it demonstrates significant methodological advances and novel biological insight into the nanoscale distribution of nuclear pore complex proteins.

The major methodological advances include:

□ Compared to other existing multicolor SD-based approaches that have been applied to biological questions Andronov et al. now demonstrate simultaneous SD-based 3-color dSTORM by using a novel triplet including CF660 with improved imaging buffer conditions. They also optimized all filters to perform 3ple color SMLM.

□ The authors also designed an integrated software package with an impressive GUI that should allow any user to easily perform SD-based SMLM. The GUI seems to include the whole workflow of drift correction, spectral-demixing, chromatic correction and filtering of weak localization.

□ The authors address the inherent problems of most previous SD-based SMLM approaches that include the inefficient usage of emitters, which was based on the avoidance of the chromatic errors due to the unevenness of the dichroic mirrors. They present a procedure for improved spectral-demixing that includes the efficient usage of all emitted photons and the correction of chromatic errors by a specialized procedure, which they call ‘splitSMLM’. However, it turns out that these improvements have already been implemented using very similar procedures in a very recent publication (Gimber et al. 2022, Nano Letters). Due to the close timing to this submission, we consider the finding here as fully independent and worth publishing despite the overlap. However, we suggest switching the focus of the manuscript more on the above-mentioned novelties. The recent publication (Gimber 2022) should, however, be cited.

A novel biological insight seems to be the side-view resolution of the NPC proteins with its NE component POM121. This shows, to our knowledge for the first time, that the POM121 is clearly distributed to the adjacent inner (not outer) membrane area of the NE using super-resolution. This might be important for experts in the field of NPC biology. This effect is clear from the selected images but it was not shown in a statistically significant manner (as many other results, s. below).

In general, beside the mentioned advances, the manuscript has major issues that should be addressed prior publication. We therefore suggest a major revision prior publication.

We thank the referee for the positive feedback and suggestions that were all taken into account in the revised manuscript as detailed below.

Specific issues:

□ In general, the manuscript (MS) is written in a rather cumbersome, wordy style and shows lots of redundancy that could be trimmed down. Also, we wonder about the structure of the MS: Figures/results are discussed and cited already in the introduction, which is rather unusual. The 'Design' section contains introductory information. Citations are not always accurately used (see below). Together, this leads to confusion and redundancy.

We have streamlined the manuscript accordingly and moved some parts to the methods section.

□ Many passages within the main text contain unclear formulations and are therefore difficult to understand. Figure legends are often cryptic and lack important information (especially the Suppl. Figures) and should be revised. We list the main passages/Figs below:

o The values of several scale-bars are missing in the legends.

Scale bar was added in Suppl. Figure S6.

o Line 64: The advantage that weak/noisy localizations can be filtered out using spectral demixing was already demonstrated (Lampe, Tadeus et al 2015; Tadeus, Lampe et al 2015). Therefore, it would be appropriate to cite those publications here for noise reduction.

Those papers are cited now.

o Line 70: Ref 16 should be added to the Refs 18-20

Done.

o Lines 86-98: Define line 92 'same channel', line 95 'narrow-band channel'

Rephrased.

o Lines 125-130: To my knowledge all previous SD-based SMLM approaches indeed used only a part of the emitted photons, not only Tadeus et al. As mentioned above, several aspects that are presented as novel improvements of the SD-based SMLM approaches have just been improved in very similar ways (Gimber 2022, s. above), which could be mentioned/discussed here. I suggest cutting down on the discussion about the chromatic errors and the usage of both emission channels, and focus the manuscript more on the above mentioned novelties (3ple color SD-dSTORM, novel GUI, biological insight).

As the Gimber 2022 work was published while our paper was in revision, we indeed were not aware of it. Thanks for informing us. We discuss it now towards the end of the manuscript.

As suggested, we have reduced the discussion part regarding chromatic errors and the detailed calculations were transferred to the methods section.

o Fig. S2: Legend too short and therefore unclear (as many other legends)

We have extended the figure legend to make it clearer.

o Fig. 1E: According to the authors the specific color assignment was done based on the univariate histogram. What criteria were used, which width of ratios included, to assign the colors. Please, explain in methods.

The univariate histogram is shown here as an example to illustrate various representations of the photon counts. In the paper we only use bivariate histograms for color assignment, as this provides more flexibility. This is now clarified in the text.

o Fig. 1F: I appreciate the efforts to explain spectral shift with/-out splitter, demonstrating the unequal chromatic distribution of photons between the channels. However, the formulation seems unclear. Please, revise. Also, the emission spectra are calculated (model) according to the methods and not (experimentally) 'observed'. Please correct. In addition, it remains unclear how the average detected wavelength (vertical dotted lines in graph) in the splitter + SplitViSu graph are calculated. Are those the mean values of the short and long wavelength graphs? If so, please, clarify within figure legend.

Those are the mean values indeed. We used the mean values because the output positions of the molecules are refined to be close to the mean positions of the coordinates from the two input channels (in order to correct for the chromatic shift). This information is now added to the figure legend.

o Fig 2A: To my understanding Fig 2A is a simulation. If so, there is no experimental data shown to support that the new procedure for multicolor registration is precise. The positive control of nanoscale 'co-localization' is missing. For example, an image of 3-color experiment of Nup96 (labeled with 3 different secondaries), in which all 3 colors should perfectly overlap.

As suggested, we have done additional experiments to address this, as discussed in the new Suppl. Fig. S5.

o Also, in line 245-247 the authors state 'The resulting localizations no noticeable chromatic shift (Fig. 2)'. Is this referring to Fig 2A?

This refers to Fig. 2A (simulation), as well as Fig. 2C-F (experimental data), where we can see that the images of the various components of the NPC are indeed coaxial plus the new Fig. S5 (new experimental data; see previous point).

o Fig. 2E, Line 245-247, Fig. S5: The images are anecdotal and lack quantification. To strengthen the finding that POM121 is more centrally distributed than Nup96, statistics on more than one experiment would be beneficial. Also, the schematic in Fig 2B could be adjusted to support this finding. In addition, the calculated FRC values in Figs. 2E and S5 are based on single ROIs. This is not representative. Please, provide statistically sound results.

Averaging of many NPCs is done now, see new panels in Fig. 2-4. The FRC values are actually based on whole images (not only on ROIs), from where various views (Fig. 2C-E, Suppl. Fig. S5) were zoomed out for visualization purposes.

o Line 452: The authors state 'absence of cross-talk'. Even though there appears to be no noticeable cross-talk within the shown images, the statement cannot be correct, as there is always some degree of cross-talk. The color cross-talk in SD-based methods is important to know and should be quantified with repeated experiments.

Using single-labelled samples, we determined the amount of cross-talk between any pair of colors to be less than 2% for the sector regions used for color separation on bivariate histograms. Hence, there is only small amount of cross-talk in our data (see new Suppl. Fig. S6-S7). We have clarified this in the MS.

o Line 453: Authors state 'absence of chromatic shifts'. The amount of remaining chromatic shift depends on the accuracy of the fit in figure S4 that is used for the correction of 'chromatic errors'. The degree of remaining shift could be illustrated with a triple color stain of one marker, see comment above (Fig. 2A).

We have done this now as discussed above; see new Suppl. Fig. S5.

o Fig. S7: axis labels missing

Corrected now.

□ The authors use both terms 'un'mixing and 'de'mixing freely in exchange. This should be standardized throughout the manuscript. Spectral 'un'mixing was originally used for linear spectral unmixing to clean up bleed through in any non-super-resolution fluorescence multi-channel method (e.g. Nadrigny et al 2006). The term 'spectral-demixing' was used specifically for the SMLM SD-dSTORM (e.g. Lampe et al 2012), as it is correctly used in the title of this manuscript. So, why not stick with 'SD' or 'spectral-demixing' as a prenoun for the SMLM method, rather than a new acronym like 'splitSMLM'? S. Line 132 and following.

We have changed “unmixing” to “demixing” everywhere in the manuscript and in the software.

Our method works for dSTORM, but not only as it extends to any SMLM method in term of data processing and optical setup. Hence, we suggest to keep our naming.

□ The authors use the term 'chromatic aberrations' for the error of localizations in both wavelength channel. The term 'chromatic aberrations' is commonly used for the focal error of lens systems (axial). For this specific case described in the manuscript, it might be better to use the term 'chromatic error' that is due to the emission splitter, which causes a lateral distortion due to nonperfect mirror flatness.

We have clarified this in the manuscript: lateral chromatic aberration (due to different spectra) and/or chromatic errors that comprise aberrations and other possible shifts between the channels due to mirrors etc.

□ *The experimental basis of shown data often remains unclear (e.g.: Fig. 1D,E). Please, provide information about the experimental statistics (i.e. number of experiments, mean value, error).*

Fig. 1D,E – this is one experiment used as an example (hence no statistics); for other experiments such as those described in Fig 2F, 3B, 4C-E, S7, S13, S15 the statistics are now indicated.

Reviewer #2:

In the manuscript “A spectral demixing method for high-precision multi-color localization microscopy applied to nuclear pore complexes” from Andronov et al, the authors describe a new method for multiplexing 2-3 colors in super resolution microscopy, utilizing an image splitter-based approach combined with a weighted algorithmic demixing. They apply this approach for three-color imaging of nuclear pore complex (NPC).

The novelty in the approach is the improved multi-color imaging that more efficiently re-distributes the photons in the 2 channels to each fluorophore. Better photon utilization translates in higher precision localization.

There are multiple commercial systems capable of performing Single molecule localization microscopy (SMLM), however multi-color SMLM remains an open challenge outside specialized labs, owing to the spectral bleed through of fluorophores and consequent incorrect assignment of photons in the super-resolution analysis. The same group has published other articles on SMLM.

Here the authors focus on analysis of 3-labeled samples, utilizing commercially available image splitters. This can be useful for other researchers in the SMLM field. In contrast to commonly used filter-based acquisition strategies, the authors propose an alternative analysis that exploits the asymmetry of fluorescence emission spectra across two spectrally separate channels to better re-assign photons to the correct fluorophores.

The authors claim that their approach:

- 1. improved localization precision & drift correction, compensation of chromatic aberrations*
- 2. optimized performance of fluorophores in a specific buffer to equalize their reactivation rates for simultaneous imaging*

Both claims are supported by the analysis of experimental data and technical specifications of the detection.

In summary, the paper presents a novel technique and, if the claims stated by the authors are confirmed (see my major comments below), it represents an original and significant technical advance in the field of super-resolution.

However, there are some fundamental points that must be addressed in order to understand the true advantages of the proposed method. The presence of instrument noise and autofluorescence is only partially discussed, and not included in the mathematical considerations. The addition of cyclooctatetraene (COT) is not fully characterized. Disadvantages or limitations of the technique are also not well described. Metrics of performance are not well reported.

Finally, the article writing requires improvement to cater for the broad audience of Nature

Communication Biology, writing can certainly be improved. The “overview” figure is not inclusive of the COT claim, which is part of the abstract. This is likely, in my opinion, to complicate understanding of the manuscript for the broad audience.

We thank the referee for the very positive feedback. We have restructured the manuscript accordingly. COT is now included in the overview figure as suggested. Instrument noise and autofluorescence are low as discussed below. We have now mentioned limitations of the method in the discussion section.

I ask the authors to address the following concerns.

Major comments

1. Page 10 “Importantly, the localization precision in this case equals that for imaging without image splitter”, this statement could be correct in an ideal, lossless optical system. In reality, however, the two camera channels (λ_S , λ_L) both cross multiple optical elements (filters, beamsplitters, mirrors). Each element has a loss (part of which is reported in Suppl. Figure 1B). As such, even if small in nature, the beam splitter introduces a loss of signal compared to a system absent of such beam splitter.

a. The authors should address this point in the main text

We now changed “equals” to “approaches”. There is some loss indeed, but it is small and much less than the ~50% that would be lost in other spectral demixing methods. We have clarified this in the revised MS (this part is now transferred to the methods section).

2. Page 13 “Spectral properties of filters for image splitting optics” The authors should utilize quantitative technical values from filters/bandpass spectra (already present in the manuscript, see Suppl. Fig. 1B) to quantify the improvement of signal utilizing their filter combination. This can be achieved by properly utilizing effectively detected area under the spectrum.

We have now done a new Suppl Fig. S10 to explain this indeed as suggested.

3. Page 13 “Very close emission spectra” (Suppl. Fig. S7) and Suppl. Fig. S7 caption “allows better separation”. The authors should quantify the improvement (x-fold or percentage) based on angle or other quantitative metrics.

We have now added angular change and r values into this figure as suggested.

4. Page 14 “detection .. through a spectral image splitter is beneficial even for single-labelled samples”, this is a valuable point of the article, as it extends the importance of the technique from a limited subset of users (multi-color SMLM) to the entire community (SMLM). This aspect in particular for identification and removal of incorrect localizations, autofluorescence and possibly background instrumental noise. Autofluorescence removal and incorrect localizations are partially described (and for autofluorescence shown in Suppl. Figure 6). However in other figures with $I_{\{L\}}$, $I_{\{S\}}$ plot the same autofluorescence is not as evident and not indicated.

a. Where is autofluorescence in Fig. 1D, Suppl. Fig. 7, Suppl. Fig. 10? The authors should address this in the manuscript.

Indeed, the method is also useful for single labelled SMLM, where it is easy to detect autofluorescence on the histogram (Supp. Figure S12). However, it is indeed difficult to identify autofluorescence in data with several fluorophores due to overlap of the autofluorescence region with that of the fluorophores' signal (this was actually discussed in lines 318-321 previous manuscript version). Therefore, we could not specifically identify autofluorescence in multi-color data. There might be still some autofluorescence that is rejected in between the selected sectors within the histograms (explained now in the MS).

b. Some $I_{\{L\}}$, $I_{\{S\}}$ plots have a selection, some other do not. This should be consistent in the manuscript.

A selection is shown when a corresponding demixed image exists for that selection (Suppl. Fig. S12), to illustrate the capabilities of the SplitViSu software (Suppl. Fig. S9) or when it is important for determination of cross-talk (new Supp. Fig. S6). In other cases, the selection is not shown to avoid overloading the figures.

c. Some selections do not fully capture all points, example Supp. Fig. 7, Supp. Fig. 10. What happens to the points not selected? How is the selection performed? The authors should address this in the manuscript.

The not selected points outside the sector are rejected and are not used for image reconstruction and other processing. The selection was performed to maximize the number of localizations while keeping cross-talk low using single-labelled samples. This is now discussed in the new results section “Cross-talk between demixed channels”, in the design part of the MS, see also new Suppl. figures S6, S7.

5. The presence of instrumental noise is evident from the spread ($I_{\{L\}}$, $I_{\{S\}}$) of the linear relation in the plots (e.g. Fig. 1D). This noise is not well discussed in the manuscript and should be part of the consideration in selecting/separating fluorophores. Authors should discuss this in the manuscript.

This is now discussed in the new results section “Cross-talk between demixed channels”. The experimental spread over the sector region describes the level of cross-talk, as now mentioned in the MS.

6. The use of cyclooctatetraene (COT) is not well integrated into the manuscript and not fully characterized. The plots in Supplementary Fig. 8 show changes in histograms of intensities but it is unclear if these changes are due to some “enhancing” photophysical effect that “increases the photon yield of all three fluorophores” or if this change is due to intrinsic signal of COT.

We have now quantified the background in presence of COT and found no increase as compared to a medium without COT (new Suppl. Fig. S15). Hence, this is an enhancing photophysical effect.

a. The authors should characterize the autofluorescence of COT (if any) and present its histogram of intensities, adding to Supp. Fig. 8 to be more convincing of this effect.

We now estimated the amount of autofluorescence and we show that there is no measurable increase due to COT (Suppl. Fig. S15). Because COT is dissolved in the buffer, it does not blink by itself, the histograms of photon counts of single-molecule localizations would be flat.

7. Page 18 “the new method significantly reduces the acquisition”. The authors should quantify this reduction in absolute or relative terms.

Up to three times for three-color samples, mentioned now in the MS.

Minor comments

1. Figure 1 does not report COT and does not have a clear outline of the proposed method. This figure can be improved.

COT is now mentioned in the figure: “sample mounted in Glox + COT”. The figure was adjusted.

2. Overall text can be improved. E.g. “COT turns out to equalize the response on reactivating”.

We have now rearranged the sections and streamlined the text.

Reviewers' comments:

Reviewer #1 (see attachment)

Reviewer #2 (Remarks to the Author):

The authors have replied to all my major comments, amended to the minor points and provided a supplementary experiments that clarify the method implementation and limitations. In the revised version, Andronov and coworkers have provided substantial improvements to the manuscript. They have now included updated clarifications in Main manuscript and Supplementary Figures, which provide a clearer description of the technology performance and capability and that will help the reader better appreciate the method.

The authors also included several experimental details that were missing in the previous version, such as characterization of the fluorophores in presence of COT . They have made considerable effort in improving clarity of description for the image processing aspect of the analysis and to characterize some technical aspects of the approach/analysis. They added clarification on the effects of on the analysis for the proposed technique. The overall effort and updated figures are commendable. I strongly recommend acceptance by Nature Communication Biology without further revision.

Point-to-point response to the reviewers' comments:

Reviewer 1 response to the author's response [only the new comments are included for simplicity]

o Line 70: Ref 16 should be added to the Refs 18-20

Done.

No, the Ref Lampe 2012 was not added to our knowledge.

Actually, it had been added, but this part was moved to line 100: “Separating fluorophores with similar spectra can be achieved with a dichroic mirror that divides the imaging path of the microscope into two channels, the short wavelength (λ_S) channel and the long wavelength (λ_L) channel, followed by simultaneous recording of the two channels on the camera^{16,25-27}”. In the current line 68, the text now discusses a different topic.

o Lines 125-130: To my knowledge all previous SD-based SMLM approaches indeed used only a part of the emitted photons, not only Tadeus et al. As mentioned above, several aspects that are presented as novel improvements of the SD-based SMLM approaches have just been improved in very similar ways (Gimber 2022, s. above), which could be mentioned/discussed here. I suggest cutting down on the discussion about the chromatic errors and the usage of both emission channels, and focus the manuscript more on the above mentioned novelties (3ple color SD-dSTORM, novel GUI, biological insight).

As the Gimber 2022 work was published while our paper was in revision, we indeed were not aware of it. Thanks for informing us. We discuss it now towards the end of the manuscript.

In terms of novelty, some sentences still have to be re-phrased to be correct.

Line 20-25 (and 80-81): ‘*So far, demixing algorithms give suboptimal results in terms of localization precision and correction of chromatic errors.*’ ... *splitSMLM ‘offers much improved localization precision & drift correction, compensation of chromatic distortions’* ...

Suggestion: “Until recently, demixing algorithms...”

Done.

About the ‘much improved resolution’ and the ‘correction of chromatic errors’, the above statement is not correct in comparison to the presented dSTORM values in Gimber et al. 2022.

Line 412 ‘*When this paper was under revision, another work was published*’

The above statement is not correct! The submission date of the paper under revision was June 27th 2022. The referred paper was published in 2022 and available as a pre-print since 2021. Please rephrase: “While this manuscript was in preparation another work”

This is incorrect. The submission date of the paper under revision was not June 27th 2022, this is the date of the revised manuscript submission. Our manuscript was submitted on March 2nd 2022.

The history is the following: Our manuscript was initially submitted to Nature Methods on December 17 2021, then it was eventually transferred to *Communications Biology* with a submission date March 2nd 2022 and the first round of revision was in March-April 2022. The Gimber *et al.* 2022 paper was published on March 15, 2022. Therefore, we believe our statement “When this paper was under revision” is correct. Our work was available as a pre-print since December 23, 2021 and the Gimber *et al.* paper since November 19, 2021. Hence, it clear that these two studies have been conducted in parallel.

As the referee initially suggested: “Due to the close timing to this submission, we consider the finding here as fully independent and worth publishing despite the overlap.”

We did in fact discuss the Gimber *et al.* work in the previous revision and even dedicate a paragraph to this in the discussion which we have further extended in the second revision, emphasizing the importance of both methods and the specific aspects of ours.

Line 69-72: “*The currently available tools for demixing^{18,20}. Current implementations of multi-color SLM with an image splitter are limited by photon budget usage that decreases localization precision¹⁸.*”

If the authors are using the word ‘current’ they would have to include Gimber et al 2022. Please, rephrase the sentences. ‘Until recently...’

Line 414/415: ‘*The procedure was applied to DNA-PAINT data using different fluorophores than those we propose in the current dSTORM work*’

Specifically, the proposed method for the ‘compensation of chromatic distortion’ has been already applied to dSTORM data in Gimber et al. 2022 (Suppl. Fig. S4), and not only to DNA-PAINT data.

The intensity-weighted multicolor registration method described in Gimber et al 2022 has been also applied to SD-dSTORM with the fluorophores Alexa647 and CF680, which have been used in this paper as well. One novelty of this paper under revision is the use of the third dye (CF660C) to achieve triple-color SD-dSTORM, which has not been demonstrated yet. This novelty could be high-lighted further.

Rephrased as suggested: removed the phrase about the same fluorophores, triple-color imaging highlighted more.

There are a couple of passages, which should be rephrased to cite the SD-DNA-PAINT paper (Gimber 2022) since it offers very similar if not identical improvements with very similar procedures.

□ “Similar to Gimber et al 2022, this method ... or here we show...”

The Gimber *et al.* paper is properly cited and discussed already.

Line 157-160;

Line 340-346; *“Thanks to simultaneous imaging and improved signal-to-noise ratios because essentially all photons are used, splitSMLM preserves the localization precision and makes drift correction much easier, which is crucial for accurate colocalization studies. The splitSMLM implementation is simple and robust and results in much improved image resolution, drift correction, compensation of chromatic errors at the nanometer scale and performance of fluorophores in optimized buffers.”*

Line 357-359: ‘The new method reduces the acquisition time...

As suggested, we have reduced the discussion part regarding chromatic errors and the detailed calculations were transferred to the methods section.

This point is not easy to verify, as we could find a few marked sections that have been transferred. The discussion is still very long, but if this appropriate for the journal, OK.

Yes, some sections have been transferred to the methods part and some paragraphs reshuffled in the previous revised version.

o Fig. S2: Legend too short and therefore unclear (as many other legends)

We have extended the figure legend to make it clearer.

S2: The channels are labeled as ‘A’ and ‘B’ in the graph, but as λ_L and λ_S channels in the legend. The legend would be easier to read if the labeling was the same.

Done in the current revised manuscript.

The new figures (averaged images and radial-/line profiles) are a very nice demonstration for the finding that POM121 is more central than NUP96.

Indeed, this is an important novel finding.

The FRC values are actually based on whole images (not only on ROIs), from where various views (Fig. 2C-E, Suppl. Fig. S5) were zoomed out for visualization purposes.

Still, the statistics should be based on repeated experiments, not only on one image. This is still a weak point within the paper. Statistics on repeated experiments would strengthen the paper.

We have now added statistics on FRC resolution values (see new **Suppl. Fig. S9**)

The increase in resolution after weighted mean procedure is apparent and logical. However, in the next step ‘refine multiple loc-s’, an averaged high-intensity loc is replaced by a cluster of single loc-s with less photons. Why do those have lower FRC values? Less photons per loc does in principle lower the resolution. Might the effect that the FRC shows a slight increase in resolution be artificial since FRC mainly relies on the density/spatial frequency of loc-s? We would appreciate a short discussion in the text.

In the step ‘refine multiple loc-s’ the averaged high-intensity localization originates from a cluster of low-intensity input localizations that have a spread of $\sigma \sim 1/\sqrt{N_{\text{low}}}$. The averaged high-intensity localization is then replaced with a cluster of localizations with a new smaller spread of $\sigma \sim 1/\sqrt{N_{\text{high}}}$, where $N_{\text{high}} = \Sigma N_{\text{low}}$. Therefore, the output localizations have smaller spread than the input localizations hence increased resolution. This is explained in the section “Refinement of multiple localizations for SMLM image reconstruction”. The localization density is indeed preserved and precision is improved as visible in the images unlike in previous methods using reduction, which lead to intensity loss (**Suppl. Fig. S8**). Therefore, the good localization density and precision provide better resolution. This is now detailed in the text.

o Line 453: Authors state ‘absence of chromatic shifts’. The amount of remaining chromatic shift depends on the accuracy of the fit in figure S4 that is used for the correction of ‘chromatic errors’. The degree of remaining shift could be illustrated with a triple color stain of one marker, see comment above (Fig. 2A).

We have done this now as discussed above; see new Suppl. Fig. S5.

Line 405,159 : ‘absence of chromatic distortions’ / ‘aberration-free’

Chromatic distortion in Fig. F5 is indeed not noticeable. However, the quality of the correction depends on the number and resolution of localizations that were used as a basis for the correction. Still ‘absence

of chromatic distortions' is wrong. I suggest to rephrase the sentence (e.g. 'no visible chromatic distortion').

Done as suggested.

- *The authors use both terms 'un'mixing and 'de'mixing freely in exchange. This should be standardized throughout the manuscript. Spectral 'un'mixing was originally used for linear spectral unmixing to clean up bleed through in any non-super-resolution fluorescence multi-channel method (e.g. Nadrigny et al 2006). The term 'spectral-demixing' was used specifically for the SMLM SD-dSTORM (e.g. Lampe et al 2012), as it is correctly used in the title of this manuscript. So, why not stick with 'SD' or 'spectraldemixing' as a prenoun for the SMLM method, rather than a new acronym like 'splitSMLM'? S. Line 132 and following.*

We have changed "unmixing" to "demixing" everywhere in the manuscript and in the software.

Our method works for dSTORM, but not only as it extends to any SMLM method in term of data processing and optical setup. Hence, we suggest to keep our naming.

The acronym 'Spectral Demixing' is not limited to dSTORM but was used for other SMLM approaches, as well (e.g. SD-DNA-PAINT). A very similar correction method to that suggested here for chromatic distortion has been applied to DNA-PAINT and dSTORM, as well (see Gimber 2022). In addition, a recent review on SMLM (Lelek et al., 2021) 'spectral demixing' is used in a broader context for a version of multicolor SMLM, suggesting that the SMLM community has adopted this term for the splitter approach.

We have now modified the text to comprise both triple-color DNA-PAINT and double-color dSTORM data.

Indeed, spectral demixing is used as a general term in the MS.

- *The experimental basis of shown data often remains unclear (e.g.: Fig. 1D,E). Please, provide information about the experimental statistics (i.e. number of experiments, mean value, error).*

Fig. 1D,E – this is one experiment used as an example (hence no statistics); for other experiments such as those described in Fig 2F, 3B, 4C-E, S7, S13, S15 the statistics are now indicated.

Fig. S13 still lacks information on statistics.

The individual data points and p-values are indicated in the histograms (now **Suppl. Fig. S14C**). Panels **Suppl. Fig. S14A-B** contain individual representative datasets.

Line 164/859: Where the FRC values ('resolution') calculated only on the shown ROI or on the whole (large overview) image? If only on ROI, please note that in the legend.

The FRC values everywhere within the manuscript were calculated on the whole images that contained one nucleus each. This is now described in the methods section and in the figure legends.

In addition, the FRC values are not calculated on repeated experiments. Please, provide statistics from data of repeated experiments. This applies also for Figs 2E, S5, S8.

We have now added statistics on FRC resolution values (see new **Suppl. Fig. S9**)

Line 77: 'excitation intensity cannot be lowered in order to improve resolution'²²,

Lower excitation would not lead to a higher, but to a lower resolution. Please correct.

Here we refer to the reference *22 Diekmann et al Nat Methods 2020* that demonstrates that reducing excitation intensity dramatically improves SMLM data quality, leading to higher photon counts per localization, higher density of localizations and higher FRC resolution.

Line 80, 337 and in general: The software splitViSu seems to be a very good integration of several established processing steps for spectral-demixing, plus a few extras (e.g. localization refinement as an alternative rendering method). However, the acronym 'splitSMLM' contains the same concept as spectral demixing including the already published 'intensity-weighted multichannel registration' procedure. Why does the paper need another acronym, instead of using 'spectral demixing'?

As outlined in the discussion, our method comprises spectral demixing for two and three fluorophores, a dedicated software implementation that comprises a full workflow convenient for users, optimally adapted emission filters, refinement of localizations and improved buffer conditions for optimized multi-colour SMLM imaging all of which are novel aspects that justify a dedicated name.

Obviously not all changes were high-lighted with the new manuscript. Therefore, we are uncertain whether we reviewed all changes.

As mentioned above, transfers to methods section and reshuffling of paragraphs were not highlighted to avoid confusion. Main text changes were highlighted throughout though.

Reviewer #2 (Remarks to the Author):

The authors have replied to all my major comments, amended to the minor points and provided a supplementary experiments that clarify the method implementation and limitations.

In the revised version, Andronov and coworkers have provided substantial improvements to the manuscript. They have now included updated clarifications in Main manuscript and Supplementary Figures, which provide a clearer description of the technology performance and capability and that will help the reader better appreciate the method.

The authors also included several experimental details that were missing in the previous version, such as characterization of the fluorophores in presence of COT. They have made considerable effort in improving clarity of description for the image processing aspect of the analysis and to characterize some technical aspects of the approach/analysis. They added clarification on the effects of on the analysis for the proposed technique.

The overall effort and updated figures are commendable.

I strongly recommend acceptance by Nature Communication Biology without further revision.

We thank the reviewer for the very positive final feedback.